# TMT-VIS: Taxonomy-aware Multi-dataset Joint Training for Video Instance Segmentation

**Rongkun Zheng**[1]    **Lu Qi**[2]    **Xi Chen**[1]    **Yi Wang**[3]
**Kun Wang**[4]    **Yu Qiao**[3]    **Hengshuang Zhao**[1*]
[1]The University of Hong Kong [2]University of California, Merced
[3]Shanghai Artificial Intelligence Laboratory [4]SenseTime Research
`{zrk22@connect, hszhao@cs}.hku.hk`

## Abstract

Training on large-scale datasets can boost the performance of video instance segmentation while the annotated datasets for VIS are hard to scale up due to the high labor cost. What we possess are numerous isolated filed-specific datasets, thus, it is appealing to jointly train models across the aggregation of datasets to enhance data volume and diversity. However, due to the heterogeneity in category space, as mask precision increases with the data volume, simply utilizing multiple datasets will dilute the attention of models on different taxonomies. Thus, increasing the data scale and enriching taxonomy space while improving classification precision is important. In this work, we analyze that providing extra taxonomy information can help models concentrate on specific taxonomy, and propose our model named **T**axonomy-aware **M**ulti-dataset Joint **T**raining for **V**ideo **I**nstance **S**egmentation (TMT-VIS) to address this vital challenge. Specifically, we design a two-stage taxonomy aggregation module that first compiles taxonomy information from input videos and then aggregates these taxonomy priors into instance queries before the transformer decoder. We conduct extensive experimental evaluations on four popular and challenging benchmarks, including YouTube-VIS 2019, YouTube-VIS 2021, OVIS, and UVO. Our model shows significant improvement over the baseline solutions, and sets new state-of-the-art records on all benchmarks. These appealing and encouraging results demonstrate the effectiveness and generality of our approach. The code is available at https://github.com/rkzheng99/TMT-VIS.

## 1 Introduction

Video Instance Segmentation (VIS) is a fundamental while challenging visual perception task that involves detecting, segmenting, and tracking object instances within videos based on a set of predefined object categories. The goal of VIS is to accurately separate the objects from the background and assign consistent instance IDs across frames, making it a crucial task for various popular downstream applications such as autonomous driving, robot navigation, video editing, etc.

Benefiting from favorable modeling of visual tokens rather than modeling pixels with CNNs, query-based transformer methods [7, 44, 16, 42, 19, 49] like Mask2Former-VIS [7], SeqFormer [44], and VITA [16] begin to dominate the VIS task. The recent success of SAM [21] demonstrates significant improvements in segmentation and detection performance through data scaling. However, mainstream VIS datasets are much smaller than widely used image detection datasets (e.g., approximately 3K videos in YouTube-VIS [47] *vs.* 120K images in COCO [27]), leaving the potential of leveraging scaled video datasets for multi-frame video input modeling largely unexplored. Previous researches [23, 10] have proved that transformer structures, when compared with CNNs, have the advantage that they can better leverage the large volume of data, and this naturally raises a question: can we alleviate this dilemma via training on large-scale datasets?

---

[*]Corresponding author

37th Conference on Neural Information Processing Systems (NeurIPS 2023).

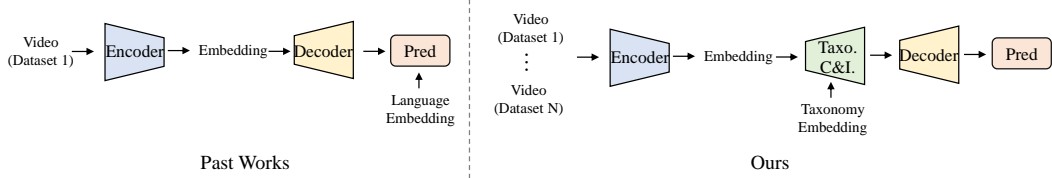

Figure 1: Comparison of multiple-dataset training paradigms. Previous multiple dataset training methods focus on unifying the label space, and some models adopt language embeddings to interact with queries after the decoder for the final classification. In order to enhance the performance, our model (right part) leverages the taxonomic embedding to refine queries through taxonomic compilation and injection (C&I) before the decoder.

Nonetheless, obtaining a single large-scale VIS dataset is cost-ineffective because of the growing need for manual labeling. A sounding alternative would be training multiple VIS datasets so as to increase data scale and enrich the taxonomy space. Still, simply combining all these datasets to train a joint model does not lead to good performance. As a result of the difference in category space, though mask precision increases with the data volume, dataset biases might hinder models from generalization: simply utilizing multiple datasets will dilute the attention of models on different categories. Therefore, increasing the data scale and enriching label space while improving classification precision become a huge challenge for researchers.

For this problem, we intend to investigate a framework that can train a unified video instance segmentor on multiple datasets while incorporating taxonomy information to boost both mask precision and classification accuracy. To this end, we propose **T**axonomy-aware **M**ulti-dataset Joint **T**raining for **V**ideo **I**nstance **S**egmentation (TMT-VIS) to jointly train multiple datasets on DETR-based methods, which can train multiple video instance segmentation datasets directly without the requirements of manually filtering irrelevant taxonomies. Built upon classic DETR-based VIS methods like Mask2Former-VIS [7], our TMT-VIS consists of two key components to tackle the problem: we first leverage a Taxonomy Compilation Module (TCM), based on a pre-trained text encoder, to compile taxonomy information from the input video. Then in the Taxonomy Injection Module (TIM), the taxonomic embeddings generated in the TCM are instilled to the visual queries in the transformer decoder through a cross-attention-based module, and thus provide a taxonomic prior to the queries. An additional taxonomy-aware matching loss is added to supervise the injection procedure. With this extra taxonomy information aggregation structure added before the transformer decoder, queries can better concentrate on the desired taxonomy in input videos. Fig. 1 demonstrates the design of our modules. By incorporating taxonomic guidance into the DETR-based model, our TMT-VIS model is able to train and utilize multiple datasets effectively.

To the best of our knowledge, TMT-VIS is the DETR-style framework that is able to jointly train multiple video instance segmentation datasets with such improvement. We evaluate our TMT-VIS on four popular VIS benchmarks, including YouTube-VIS 2019 and 2021 [47], OVIS [35], and UVO [39]. The extensive experimental evaluations show the effectiveness and generality of our method with significant improvement over the baselines. For example, compared with Mask2Former-VIS [7] with the ResNet-50 backbone, our TMT-VIS gets absolute AP improvements of 3.3%, 4.3%, 5.8%, and 3.5% on the aforementioned challenging benchmarks, respectively. Compared with another high-performance solution VITA [16], our solution gets absolute AP improvements of 2.8%, 2.6%, 5.5%, and 3.1%, respectively. Finally, the proposed TMT-VIS sets new state-of-the-art records on these popular and challenging benchmarks. Our main contributions can be summarized threefold:

- We analyze the limitations of existing video instance segmentation methods and propose a novel multiple-dataset training algorithm named TMT-VIS, which can well utilize taxonomic guidance to train and utilize multiple video instance segmentation datasets effectively.

- We develop a two-stage module: Taxonomy Compilation Module (TCM) and Taxonomy Injection Module (TIM). The former adopts the CLIP text encoder to compile taxonomic information in videos, while the latter utilizes the taxonomic information in TCM to inject guidance into queries.

- We conduct extensive experimental evaluations on four popular and challenging VIS benchmarks, including YouTube-VIS 2019, YouTube-VIS 2021, OVIS, and UVO, and the achieved significant improvements demonstrate the effectiveness and generality of the proposed approach.

## 2   Related Work

**Video instance segmentation.** The current VIS methods can be categorized into two: online and offline. Online VIS methods typically adopt a tracking-by-detection approach. They first predict object detection and segmentation results within several frames, and link the outputs with the same identities across frames using optimized matching algorithms, which can be either hand-crafted or learnable-based. MaskTrack R-CNN [47] is the baseline online VIS method, which is built by adding a simple tracking branch to Mask R-CNN [13]. Following works [5, 48, 28, 18] adopt the similar motivation, measuring the similarities between frame-level predictions and associating them through different matching module designs. Inspired by Video Object Segmentation [33] ,Multi-Object Tracking (MOT) [9], and Multi-Object Tracking and Segmentation (MOTS) [37], recent VIS works such as [11, 26, 12] are proposed. While many MOT methods use multiple trackers [2, 34, 50], some more recent MOT works [53, 30] utilize trajectory queries to exploit inter-instance relationships. Motivated by such approaches, GenVIS [15] implements a novel target label assignment and instance prototype memory in query-based sequential learning. On the other hand, IDOL [45] introduces a contrastive learning head for discriminative instance embeddings, based on Deformable-DETR [54].

In contrast to online methods, offline methods aim to predict masks and trajectories of instances in the entire video in one step by feeding the entire video. STEm-Seg [1] utilizes a single-stage model that gradually updates and clusters spatio-temporal embeddings. MaskProp [3] and Propose-Reduce [26] improve mask quality and instance relations through mask propagation. VisTR [42] successfully adapts DETR [6] to the VIS task for the first time, which adopts instance queries to model the whole video. Mask2Former-VIS [7], which is an adaptation of Mask2Former for 3D spatio-temporal features, has become the leading method due to its mask-focused representation. Not long ago, VITA has emerged, as detailed in [16], which employs object tokens to model the entire video and uses video queries to decode object information from these tokens.

**Multi-dataset training.** It is widely acknowledged that leveraging large-scale data in visual recognition tasks can notably improve performance [4, 31, 52, 51, 40, 36], but collecting annotated data is challenging to scale. In object detection, several recent studies have explored using multiple datasets to train a single detector and improve feature representations. For example, BigDetection [4] merges several datasets with a hand-crafted label mapping in a unified label space. Detection Hub [31] unifies multiple datasets by adapting queries on category embeddings, considering each dataset's category distributions. UniDet [52] trains a unified detector with split classifiers and incorporates a cost for merging categories across datasets to automatically optimize for a common taxonomy. OmDet [51] uses a language-based framework to learn from multiple datasets, but its specific architecture limits the transferability of pre-trained models to other detectors. In video tasks such as action recognition, several works [32, 17, 38, 22, 41] propose to combine multiple video datasets for training. PolyViT [24] further extends this approach by including image, video, and audio datasets using different sampling procedures. MultiTrain [23] utilizes informative representation regularizations and projection losses to learn robust and informative feature representations from multiple action recognition datasets. Despite achieving huge progress, previous language-based works simply use language embeddings in the end to match with outputs in order to get class predictions, whereas our solution leverages the taxonomic embedding to refine queries before the decoder which helps queries to concentrate on desired taxonomy, resulting in a huge improvement in performance.

The UNINEXT[25] is the first DETR-based method that jointly trains multiple VIS datasets. It simply utilizes the BERT language encoder to generate language embeddings of categories from all video datasets, and fuses the information with visual embeddings through a simple bi-directional cross-attention module. However, UNINEXT has no video-specific design, and it doesn't use the language embeddings to predict a set of possible set of categories, so the semantic information of VIS categories is simply aggregated without further operations.

## 3   Method

In this section, we will first introduce the overall architecture of the proposed framework **T**axonomy-aware **M**ulti-dataset Joint **T**raining for **V**ideo **I**nstance **S**egmentation (TMT-VIS) in Sec. 3.1. Then, we will detail the exquisitely designed taxonomy aggregation module, including Taxonomy Compilation Module (TCM) in Sec. 3.2 and Taxonomy Injection Module in Sec. 3.3. Following this, we further

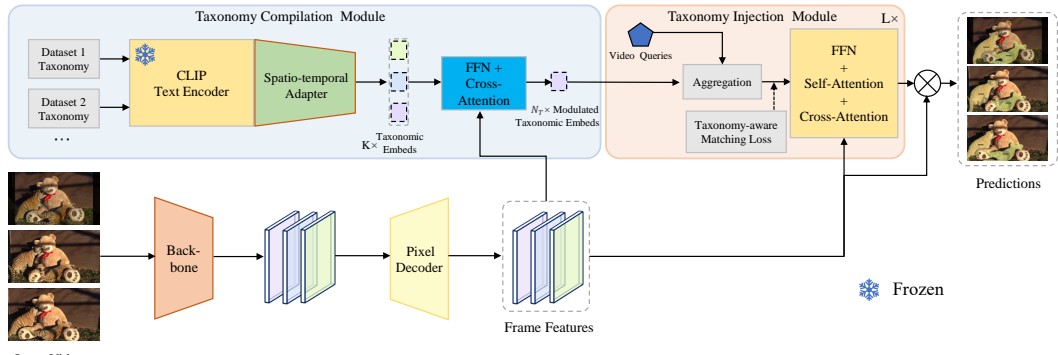

Figure 2: Overall framework of the proposed TMT-VIS method. Taxonomy Compilation Module (TCM) adopts the CLIP text encoder and spatio-temporal Adapter to generate video-specific modulated taxonomic embeddings. Taxonomy Injection Module (TIM) leverages the modulated embeddings to provide taxonomic guidance to visual queries in the decoder. An additional taxonomy-aware loss is added to supervise the compilation.

demonstrate the taxonomy-aware matching loss and total optimization target in Sec. 3.4. Finally, we give several implementation details of the framework in Sec. 3.5.

## 3.1 Overall architecture.

Built upon a DETR [6] style framework Mask2Former [8], we inherit the formulation that treats video instance segmentation as a set prediction problem. To enhance classification performance, we incorporate unified category information as an additional input:

$$\mathbf{E} = f(\mathbf{V}, \mathbf{Q}|f_{\text{text}}(\mathbf{L})), \tag{1}$$

where $f$ stands for our method and $f_{\text{text}}$ is a pretrained text encoder. $\mathbf{V}$ is an input video represented as an image sequence, $\mathbf{L}$ is the label space (referred to as taxonomy), and $\mathbf{E}$ is the embedding to predict $N$ segmentation masks, $\mathbf{Q}$ referring to queries.

With the improved formulation, we introduce a new framework TMT-VIS, which incorporates taxonomy information into queries to improve the classification ability of the model trained on multiple datasets. As shown in Fig. 2, the framework contains two modules: Taxonomy Compilation Module (TCM) and Taxonomy Injection Module (TIM). By interacting video features with adapted taxonomic embedding, we generate video-specific taxonomic prior. Then, we integrate the learned taxonomic embeddings with the learnable queries, enabling better semantic representation through a divide-and-conquer strategy. We will discuss these key modules, after providing a brief overview of the Mask2Former used.

**Revisiting Mask2Former.** Mask2Former [7, 8] is a universal Transformer-based framework for image or video instance segmentation. Mask2Former implements a Transformer-based decoder, which recursively learns and refines $N$ number of $C$-dimensional queries $\mathbf{Q} \in \mathbb{R}^{N \times C}$ to generate embeddings $\mathbf{E}$. Specifically, the transformer decoder is a nine-layer structure, where $l^{\text{th}}$ layer contains a masked cross-attention, a self-attention, and a feed-forward network in a cascaded manner.

## 3.2 Taxonomy Compilation Module

For joint-training multiple datasets, the biggest challenge is label inconsistency, where taxonomy from different datasets differs. Given multiple annotated VIS datasets $\mathcal{D} = \{\mathcal{D}_1, \mathcal{D}_2, \ldots, \mathcal{D}_n\}$, we have the whole label space $\mathcal{L} = \{\mathcal{L}_1, \mathcal{L}_2, \ldots, \mathcal{L}_n\}$. In our Taxonomy Compilation Module (TCM), we utilize the text encoder of the CLIP to generate taxonomy embedding $\mathcal{E} = \{\mathcal{E}_i, |i = 1, \ldots, K\}$, where $K$ is the total number of category in $\mathcal{L}$, with $\mathcal{E}_i \in \mathbb{R}^d$ representing the text embedding of a specific category, where $d$ is the dimension size of the embedding. When given a video with $T$ frames as input, we have the video features $\mathcal{F} \in \mathbb{R}^{HW \times D \times T}$ extracted by the Mask2Former-VIS's Pixel Decoder, where $H$ and $W$ represent the height and weight of the feature map. To align the dimensions between text embeddings and video features, we add a Spatio-temporal Adapter after the text encoder

of the CLIP. Concretely, we use two FC layers and an activation layer in the middle, with the first layer projecting the original embedding to a lower dimension and the second one projecting to align with the dimension of video features. The adapted text embeddings $\mathcal{E}^1$ are sent into cross-attention and FFN operations. We gradually update the taxonomic embeddings layer by layer by aggregating context information from the input video features via cross-attention and FFN, in order to unearth the potential taxonomy contained in images. After this, we calculated the dot product between the different modulated taxonomic embeddings to predict the most relevant taxonomy in the given video according to the score $\mathcal{S}$. Note that $\cdot$ indicates the dot production.

The whole process of TCM can be formulated as follows:

$$\mathcal{E}^1 = \text{Adapter}(\mathcal{E}), \tag{2}$$

$$\mathcal{E}^2 = \text{FFN}(\text{CrossAttn}(\mathcal{E}^1, \mathcal{F})), \tag{3}$$

$$\mathcal{E}^3 = \text{FFN}(\text{CrossAttn}(\mathcal{E}^2, \mathcal{F})), \tag{4}$$

$$\mathcal{S} = \text{Sigmoid}(\text{Linear}(\mathcal{E}^3) \cdot \mathcal{E}^1). \tag{5}$$

Thus, we can update the taxonomy set by selecting the top $N_T$ embeddings with higher scores in $\mathcal{S}$. This suggests that there are $N_T$ corresponding categories that are most likely contained in the input video. The compiled taxonomic embeddings $\mathcal{E}_\mathcal{T}$ by concatenating the $N_T$ embeddings from $\mathcal{E}^3$.

### 3.3 Taxonomy Injection Module

In DETR-based methods, learnable queries in the decoder are usually zero-initialized, which is lack of taxonomy information. With the supervision of taxonomy priors, query features are more likely to converge faster and predict the desired categories. Also, since utilizing multiple datasets will dilute the attention of models on different categories, the aggregation of taxonomic embeddings can help query features concentrate on the predicted subset of taxonomies. Thus, in Taxonomy Injection Module (TIM), we attempt to incorporate the extracted embeddings from TCM to the queries in the decoder structure. The process of TIM can be formulated as follows:

$$\mathbf{X}'_{l-1} = \text{FFN}(\text{CrossAttn}(\mathbf{X}_{l-1}, \mathcal{E}_T)), \tag{6}$$

$$\mathbf{X}_l = \text{FFN}(\text{SelfAttn}(\text{CrossAttn}(\mathbf{X}'_{l-1}, \mathcal{F}))). \tag{7}$$

where $\mathbf{X}_l$ represents the query features of the $l^{th}$ layer of the transformer decoder, $l \in [1, L], L = 9$.

By aggregating $\mathcal{E}_T$ into query features through cross-attention and self-attention modules, we could have the refined query features with richer taxonomy information, and thus filtering irrelevant background taxonomy information contained in $\mathbf{X}_l$. This strategy shares the same motivation with the masked attention in Mask2Former, which tries to mask out background regions in the spatial dimension, whereas our method alternatively masks out irrelevant taxonomies by using the modulated taxonomic embeddings $\mathcal{E}_T$.

### 3.4 Taxonomy-aware Matching Loss

Typically, DETR-like models view VIS as a set prediction problem and they rely on global one-to-one assignment for bipartite matching, and the Hungarian algorithm is employed to find an optimal assignment with all categories. The classification loss is based on the results of the matching. Inspired by this matching, as we utilize taxonomic embeddings as guidance to refine instance queries, we introduce an extra loss $\mathcal{L}_{taxo}$ to supervise the injection of taxonomic embeddings. Specifically, we add a text-aware matching cost: the matching algorithm remains the same, which is cross-entropy loss, but the predicted class are not from the final prediction, but from the prediction right after the injection. By adding this supervision, we further guarantee the guidance provided by the compiled taxonomic embeddings is successfully injected. Finally, we integrate all losses together as follows:

$$\mathcal{L} = \mathcal{L}_{\text{mask}} + \lambda_{\text{cls}}\mathcal{L}_{\text{cls}} + \lambda_{\text{taxo}}\mathcal{L}_{\text{taxo}}. \tag{8}$$

and we set $\lambda_{\text{cls}} = 2.0$ and $\lambda_{\text{taxo}} = 0.5$.

### 3.5 Implementation Details

Our method is implemented on top of detectron2 [46]. Hyper-parameters regarding the pixel decoder and transformer decoder are consistent with the settings of Mask2Former-VIS [7]. In the Taxonomy

Table 1: Results comparison on the YouTube-VIS 2019 and 2021 validation sets. We group the results by online or offline methods, and then with ResNet-50 or Swin-L backbone structures. TMT-VIS is the model we built upon Mask2Former-VIS, while TMT-VIS$^\dagger$ in 'Online' and 'Offline' are the model that we add our designs based on popular online and offline method, GenVIS and VITA. Our algorithm gets the highest AP performance compared with recent approaches.

| Method | Backbone | YouTube-VIS 2019 | | | | | YouTube-VIS 2021 | | | | |
|---|---|---|---|---|---|---|---|---|---|---|---|
| | | AP | $AP_{50}$ | $AP_{75}$ | $AR_1$ | $AR_{10}$ | AP | $AP_{50}$ | $AP_{75}$ | $AR_1$ | $AR_{10}$ |
| MaskTrack R-CNN [47] | ResNet-50 | 38.6 | 56.3 | 43.7 | 35.7 | 42.5 | 36.9 | 54.7 | 40.2 | 30.6 | 40.9 |
| CrossVIS [48] | ResNet-50 | 36.3 | 56.8 | 38.9 | 35.6 | 40.7 | 34.2 | 54.4 | 37.9 | 30.4 | 38.2 |
| MinVIS [18] | ResNet-50 | 47.4 | 69.0 | 52.1 | 45.7 | 55.7 | 44.2 | 66.0 | 48.1 | 39.2 | 51.7 |
| IDOL [45] | ResNet-50 | 49.5 | 74.0 | 52.9 | 47.7 | 58.7 | 43.9 | 68.0 | 49.6 | 38.0 | 50.9 |
| GenVIS [15] | ResNet-50 | 51.3 | 72.0 | 57.8 | 49.5 | 60.0 | 46.3 | 67.0 | 50.2 | 40.6 | 53.2 |
| **TMT-VIS$^\dagger$** | ResNet-50 | **53.9** | **74.8** | **59.1** | **51.4** | **62.7** | **49.4** | **69.1** | **51.8** | **43.6** | **54.8** |
| MinVIS [18] | Swin-L | 61.6 | 83.3 | 68.6 | 54.8 | 66.6 | 55.3 | 76.6 | 62.0 | 45.9 | 60.8 |
| IDOL [45] | Swin-L | 64.3 | 87.5 | 71.0 | 55.6 | 69.1 | 56.1 | 80.8 | 63.5 | 45.0 | 60.1 |
| GenVIS [15] | Swin-L | 63.8 | 85.7 | 68.5 | 56.3 | 68.4 | 60.1 | 80.9 | 66.5 | 49.1 | 64.7 |
| **TMT-VIS$^\dagger$** | Swin-L | **65.4** | **88.2** | **72.1** | **56.4** | **69.3** | **61.9** | **82.0** | **68.3** | **51.2** | **65.9** |
| EfficientVIS [43] | ResNet-50 | 37.9 | 59.7 | 43.0 | 40.3 | 46.6 | 34.0 | 57.5 | 37.3 | 33.8 | 42.5 |
| IFC [19] | ResNet-50 | 41.2 | 65.1 | 44.6 | 42.3 | 49.6 | 35.2 | 55.9 | 37.7 | 32.6 | 42.9 |
| Mask2Former-VIS [7] | ResNet-50 | 46.4 | 68.0 | 50.0 | - | - | 40.6 | 60.9 | 41.8 | - | - |
| TeViT [49] | MsgShifT | 46.6 | 71.3 | 51.6 | 44.9 | 54.3 | 37.9 | 61.2 | 42.1 | 35.1 | 44.6 |
| SeqFormer [44] | ResNet-50 | 47.4 | 69.8 | 51.8 | 45.5 | 54.8 | 40.5 | 62.4 | 43.7 | 36.1 | 48.1 |
| VITA [16] | ResNet-50 | 49.8 | 72.6 | 54.5 | 49.4 | 61.0 | 45.7 | 67.4 | 49.5 | 40.9 | 53.6 |
| **TMT-VIS** | ResNet-50 | 49.7 | 73.4 | 53.9 | 49.2 | 60.7 | 44.9 | 66.1 | 48.5 | 39.8 | 52.1 |
| **TMT-VIS$^\dagger$** | ResNet-50 | **52.6** | **74.4** | **57.3** | **50.6** | **61.8** | **48.3** | **69.8** | **50.8** | **42.0** | **55.2** |
| SeqFormer [44] | Swin-L | 59.3 | 82.1 | 66.4 | 51.7 | 64.4 | 51.8 | 74.6 | 58.2 | 42.8 | 58.1 |
| Mask2Former-VIS [7] | Swin-L | 60.4 | 84.4 | 67.0 | - | - | 52.6 | 76.4 | 57.2 | - | - |
| VITA [16] | Swin-L | 63.0 | **86.9** | 67.9 | 56.3 | 68.1 | 57.5 | 80.6 | 61.0 | 47.7 | 62.6 |
| **TMT-VIS** | Swin-L | 63.2 | 85.2 | 68.3 | 56.4 | 68.2 | 56.4 | 80.2 | 61.3 | 47.0 | 61.4 |
| **TMT-VIS$^\dagger$** | Swin-L | **64.9** | 86.1 | **69.7** | **57.2** | **69.3** | **59.3** | **81.0** | **63.7** | **48.9** | **63.5** |

Compilation Module, the size of the taxonomy embedding set $N_T$ is set to 10, which matches the maximum instance number per video. Referring to Mask2Former-VIS [7], we initially train it on the COCO [27] dataset before fine-tuning it on VIS datasets. During inference, we resize the shorter side of each frame to 360 pixels for ResNet [14] backbones and 480 pixels for Swin [29] backbones. In the inference part, our SOTA performance method utilizes the given vocabulary of the dataset. However, our method could still perform well with a unified vocabulary, which is credited to the utilization of the CLIP encoder.

## 4 Experiments

In this part, we first give some details about the experimented benchmarks in Sec. 4.1. Then, we report the main results in Sec. 4.2, which mainly demonstrates the superiority of the proposed approach over previous solutions. Furthermore, we give detailed ablation studies in Sec. 4.3 and Sec. 4.4 to validate the exquisite design of the whole framework and related proposed modules. Finally, we illustrate some qualitative visualizations in Sec. 4.5 to show the effectiveness.

### 4.1 Datasets

We conduct extensive experimental evaluations on four popular and challenging benchmarks, including YouTube-VIS 2019 and 2021 [47], OVIS [35], and UVO [39]. Key statistics of these datasets are presented in the Appendix. YouTube-VIS 2019 [47] is the first large-scale dataset for video instance segmentation, with 2.9K videos averaging 4.61s in duration and 27.4 frames in validation videos. YouTube-VIS 2021 [47] includes more challenging longer videos with more complex trajectories, resulting in an average of 39.7 frames in validation videos. OVIS [35] dataset is another challenging VIS dataset, offering 25 object categories and focusing on complex scenes with significant occlusions between objects. While only containing 607 training videos, its videos are much longer, lasting

Table 2: Results comparison on the UVO and OVIS validation sets. We group the results by online or offline methods, and then with ResNet-50 or Swin-L backbone structures. TMT-VIS is the model built upon Mask2Former-VIS, while TMT-VIS$^\dagger$ in 'Online' and 'Offline' are the model that we add our designs based on popular online and offline method, GenVIS and VITA. Our algorithm gets the highest AP performance compared with recent approaches.

| | Method | Backbone | OVIS | | | | | UVO | | |
| | | | AP | $AP_{50}$ | $AP_{75}$ | $AR_1$ | $AR_{10}$ | AP | $AP_{50}$ | $AP_{75}$ |
|---|---|---|---|---|---|---|---|---|---|---|
| Online | MaskTrack R-CNN [47] | ResNet-50 | 10.8 | 25.3 | 8.5 | 7.9 | 14.9 | 9.3 | 20.9 | 8.2 |
| | CrossVIS [48] | ResNet-50 | 14.9 | 32.7 | 12.1 | 10.3 | 19.8 | - | - | - |
| | MinVIS [18] | ResNet-50 | 25.0 | 45.5 | 24.0 | 13.9 | 29.7 | - | - | - |
| | IDOL [45] | ResNet-50 | 30.2 | 51.3 | 30.0 | 15.0 | 37.5 | 16.8 | 28.1 | 17.3 |
| | GenVIS [15] | ResNet-50 | 35.8 | 60.8 | 36.2 | 16.3 | 39.6 | - | - | - |
| | **TMT-VIS$^\dagger$** | ResNet-50 | **38.4** | **62.1** | **39.5** | **17.3** | **41.5** | - | - | - |
| | MinVIS [18] | Swin-L | 39.4 | 61.5 | 41.3 | 18.1 | 43.3 | - | - | - |
| | IDOL [45] | Swin-L | 42.6 | 65.7 | 45.2 | 17.9 | 49.6 | - | - | - |
| | GenVIS [15] | Swin-L | 45.2 | 69.1 | 48.4 | **19.1** | 48.6 | - | - | - |
| | **TMT-VIS$^\dagger$** | Swin-L | **46.9** | **71.0** | **48.9** | 18.8 | **52.0** | - | - | - |
| Offline | IFC [19] | ResNet-50 | 13.1 | 27.8 | 11.6 | 9.4 | 23.9 | - | - | - |
| | Mask2Former-VIS [7] | ResNet-50 | 16.5 | 36.5 | 14.6 | 10.2 | 23.4 | 18.2 | 29.7 | 18.9 |
| | SeqFormer [44] | ResNet-50 | 15.1 | 31.9 | 13.8 | 10.4 | 27.1 | - | - | - |
| | VITA [16] | ResNet-50 | 19.6 | 41.2 | 17.4 | 11.7 | 26.0 | 18.9 | 30.6 | 19.8 |
| | **TMT-VIS** | ResNet-50 | 22.8 | 43.6 | 21.7 | 13.1 | 28.3 | 21.2 | 32.9 | 22.1 |
| | **TMT-VIS$^\dagger$** | ResNet-50 | **25.1** | **45.9** | **23.8** | **14.2** | **29.9** | **22.0** | **33.4** | **22.9** |
| | Mask2Former-VIS [7] | Swin-L | 23.1 | 45.4 | 21.8 | 13.3 | 29.2 | 27.3 | 42.0 | 27.2 |
| | VITA [16] | Swin-L | 27.7 | 51.9 | 24.9 | 14.9 | 33.0 | - | - | - |
| | **TMT-VIS** | Swin-L | 28.0 | 50.9 | 27.6 | 14.4 | 34.1 | 29.0 | 42.8 | 29.5 |
| | **TMT-VIS$^\dagger$** | Swin-L | **32.5** | **55.1** | **32.0** | **15.4** | **38.3** | **29.9** | **43.6** | **30.1** |

12.77s on average. UVO [39] dataset has 81 object categories (80 shared with COCO [27], plus an extra "other" category for non-COCO instances). UVO, a subset of 1.2K Kinetics-400 [20] videos, contains 503 videos densely annotated at 30fps, and provides segmentation masks exhaustively for all object instances.

## 4.2 Main Results

We compare TMT-VIS with state-of-the-art approaches which are with ResNet-50 and Swin-L backbones on the YouTube-VIS 2019 and 2021 [47], OVIS [35], and UVO [39] benchmarks. The results are reported in Tables 1 and 2.

**YouTube-VIS 2019.** Table 1 shows the comparison on YouTube-VIS 2019 dataset. Our TMT-VIS sets new state-of-the-art results under all of the settings. Based on Mask2Former-VIS, TMT-VIS gets 49.7% AP and 63.2% with ResNet-50 and Swin-L backbones, respectively, outperforming the baseline by 3.3% and 2.8% absolutely. When the proposed designs are added onto VITA, TMT-VIS gets 52.6% AP and 64.9% AP with ResNet-50 and Swin-L backbones, respectively, outperforming the baseline by 2.8% and 1.9%. When GenVIS is added with our design, TMT-VIS gets 53.9% AP and 65.4% AP with ResNet-50 and Swin-L backbones, outperforming the baseline by 2.6% and 1.6%.. We list the model parameters of Mask2Former (216M), SeqFormer (220M), VITA (247M), and our TMT-VIS (255M), which shows that TMT-VIS performs notably better with similar model parameters. Also, we compared our TMT-VIS with UNINEXT [25] on YouTube-VIS 2019 dataset. With our delicate design of TCM & TIM, our method achieves a higher performance of 64.9% AP with 4 VIS datasets on Swin-L backbone, and the total training time is approximately 1 day 12 hours. On the other hand, UNINEXT is trained on 8 video datasets and the total training time is

Table 3: Ablation study on training with multiple VIS datasets with Mask2Former-VIS (which is abbreviated as 'M2F') and TMT-VIS and their validation results on various VIS datasets. Experiments 1 - 7 report the results of training with M2F. Experiments 8 - 14 report the results of training with TMT-VIS .

| ID | Method | $YTVIS_{train}$ | $OVIS_{train}$ | $UVO_{train}$ | $YTVIS_{val}$ | $OVIS_{val}$ | $UVO_{val}$ |
|---|---|---|---|---|---|---|---|
| 1 | M2F | ✓ | | | 46.4 | 2.3 | 1.9 |
| 2 | M2F | | ✓ | | 5.2 | 16.5 | 3.6 |
| 3 | M2F | | | ✓ | 4.4 | 2.5 | 18.2 |
| 4 | M2F | ✓ | ✓ | | 47.3 | 17.4 | 4.9 |
| 5 | M2F | ✓ | | ✓ | 46.2 | 3.7 | 19.0 |
| 6 | M2F | | ✓ | ✓ | 7.1 | 16.6 | 18.7 |
| 7 | M2F | ✓ | ✓ | ✓ | 7.1 | 16.6 | 18.7 |
| 8 | TMT-VIS | ✓ | | | 47.3 | 7.2 | 6.5 |
| 9 | TMT-VIS | | ✓ | | 10.5 | 17.8 | 8.0 |
| 10 | TMT-VIS | | | ✓ | 10.1 | 8.6 | 18.8 |
| 11 | TMT-VIS | ✓ | ✓ | | 48.8 | 20.9 | 10.1 |
| 12 | TMT-VIS | ✓ | | ✓ | 47.0 | 10.3 | 20.4 |
| 13 | TMT-VIS | | ✓ | ✓ | 14.8 | 19.4 | 20.2 |
| 14 | TMT-VIS | ✓ | ✓ | ✓ | 49.7 | 22.8 | 21.2 |

Table 4: Ablation study on key component designs in TMT-VIS. The 'Multiple Datasets' setting refers to training on the three VIS datasets (Youtube-VIS 2019, OVIS, and UVO). Experiment I refers to the Mask2Former-VIS model. Note that all experiments are evaluated on Youtube-VIS 2019, abbreviated as 'YTVIS'.

| ID | TCM&TIM | Taxonomy loss | YTVIS | | | Multiple Datasets | | |
|---|---|---|---|---|---|---|---|---|
| | | | AP | $AP_{50}$ | $AP_{75}$ | AP | $AP_{50}$ | $AP_{75}$ |
| I | | | 46.4 | 68.0 | 50.0 | 47.2 | 69.1 | 50.7 |
| II | ✓ | | 47.0 | **69.1** | 50.3 | 49.2 | 72.7 | 53.4 |
| III | ✓ | ✓ | **47.3** | 68.9 | **50.8** | **49.7** | **73.4** | **53.9** |

approximately 3 days, but its performance is 64.3% AP. Our method is proved to be more effective and data-efficient.

**YouTube-VIS 2021.** Table 1 also compares the results on the YouTube-VIS 2021 dataset. Specifically, based on Mask2Former-VIS, TMT-VIS achieves 44.9% AP and 56.4% AP with ResNet-50 and Swin-L backbones, respectively. Compared to the baseline, TMT-VIS boosts performance by 4.3 and 3.8 points. When the exquisite designs are added to VITA, TMT-VIS achieves 48.3% AP and 59.3% AP respectively, boosting baseline performance by 2.6 and 1.8 points. When plugging to GenVIS, TMT-VIS achieves 49.4% AP and 61.9% AP respectively, boosting baseline performance by 3.1% AP and 1.8% AP.

**OVIS and UVO.** Table 2 illustrates the competitiveness of TMT-VIS on the challenging OVIS dataset. Particularly, TMT-VIS achieves 25.1% AP and 32.5% AP with ResNet-50 and Swin-L backbones with VITA as the base model, and harvests 38.4% AP and 46.9% AP with GenVIS as the base model on these backbones. Table 2 shows the strong performance of TMT-VIS among offline methods on the challenging UVO dataset. These appealing and encouraging results further prove the effectiveness and generality of the proposed approach on multiple dataset training.

### 4.3 Ablation Studies

In the following, we conduct extensive ablation studies to check the properties of each component. We first show that our algorithm can improve model performance by a large margin when trained on different combinations of datasets. Then we examine the detailed ablations of our design. Note that all of the TMT-VIS ablation experiments are conducted based on Mask2Former-VIS with the ResNet-50 backbone.

**Multiple dataset joint-training problem.** Even though mask precision increases with the data volume, due to the heterogeneity in category space, simply utilizing multiple datasets will dilute

Table 5: Ablation study on sampling ratio of multiple datasets (The order of the datasets is YTVIS:UVO:OVIS).

| Sampling Ratio | AP | AP$_{50}$ | AP$_{75}$ |
|---|---|---|---|
| 2:1:0.75 | 48.6 | 71.0 | 52.4 |
| 1:1:0.5 | 49.4 | 72.8 | 53.2 |
| 1:1:0.75 | **49.7** | **73.4** | **53.9** |
| 1:1:1 | 49.5 | 72.6 | 53.7 |

Table 6: Ablation study on the size of taxonomic embedding set in TCM.

| Size | AP | Size | AP |
|---|---|---|---|
| 1 | 47.1 | 15 | 49.2 |
| 2 | 47.5 | 20 | 48.5 |
| 5 | 49.3 | 50 | 47.4 |
| 10 | **49.7** | 100 | 47.2 |

Table 7: Ablation study on aggregation strategy in TIM.

| Method | AP | AP$_{50}$ | AP$_{75}$ |
|---|---|---|---|
| Add | 47.6 | 71.8 | 50.3 |
| Concatenation | 47.4 | 71.0 | 51.1 |
| Cross-attention | **49.7** | **73.4** | **53.9** |

the attention of models on different categories. In Table 3, we carefully examine the performance of multiple datasets joint training on various datasets. We find that the popular Mask2Former-VIS framework meets difficulties in dealing with datasets with different taxonomy spaces. As the number of datasets increases, the performance may even decrease. In contrast, as we increase the number of datasets, the performance of our TMT-VIS improves gradually.

**Key component designs.** Table 4 demonstrates the effect of our component designs when combined with the prevalent Mask2Former-VIS. By adopting our algorithm, Mask2Former-VIS achieves a huge gain of 2.5 points in AP performance when both models are trained with three VIS datasets. Our extra loss also demonstrates a gain of 0.5 AP.

**Datasets sampling ratio.** The sampling ratio of different VIS datasets is a crucial parameter. Diverse sampling ratios can cause model to focus on different categories, which can impact the attention and performance of the model. In this study, we arrange the sampling ratio according to the optimal number of iterations each dataset has trained individually. As shown in Table 5, '1:1:0.75' ratio yields the best results, which aligns with our previous training approaches on a single dataset. The performance gap between different ratios validates that selecting an appropriate sampling ratio is important for ensuring that the model learns the relevant category information effectively and achieves optimal performance during multi-dataset training.

## 4.4   TCM and TIM

In Tables 6 and 7, we investigate the effect of different sizes of taxonomic embedding set as well as the aggregation strategy used in TCM and TIM.

**Size of taxonomic embedding set.** Table 6 displays the performance of TMT-VIS with varying numbers of output taxonomic embedding $N_T$ in the TCM from 1 to 100. The results indicate that the model performs best when $N_T = 10$ embeddings are selected as input embeddings to TIM. When $N_T$ is set to 1 or 2, which is apparently less than the overall categories in input video, topk taxonomic embeddings miss the groundtruth classes. As $N_T$ decreases, the aggregated taxonomic guidance contained in embeddings becomes insufficient, and the selected embeddings may not encode taxonomic semantics for all instances in the video, resulting in a drop in performance. Conversely, when $N_T$ gets larger than the optimum value, the redundant taxonomic embeddings dilute the category information, leading to performance degradation. When using all embeddings, the improvement is trivial since TCM can not provide filtration to irrelevant classes, and the TIM module is simply injecting the information of the whole category space to queries.

**Aggregation strategy.** Table 7 demonstrates the results of different aggregation strategies implemented in the TIM. There are multiple ways of aggregating taxonomic embeddings to queries in the transformer decoder: in the 'add' setting, taxonomic embeddings are expanded to match the dimensions of queries and addition is applied to the inflated embeddings; in the 'concatenation' setting, taxonomic embeddings are concatenated with queries to provide additional semantic queries; in the 'cross-attention' setting, taxonomic embeddings are fed to the cross-attention module as key and values to inject category semantics into queries. The results indicate that aggregating taxonomy guidance can improve multi-dataset training performance, and that cross-attention is found to be the most effective strategy for injecting taxonomic embeddings into instance queries. On the other hand, directly concatenating or adding taxonomic embeddings does not improve significantly. This suggests that straightforward modifications to instance queries don't infuse taxonomy information into queries, and such superficial modifications will be refined after passing the cascades of the decoder.

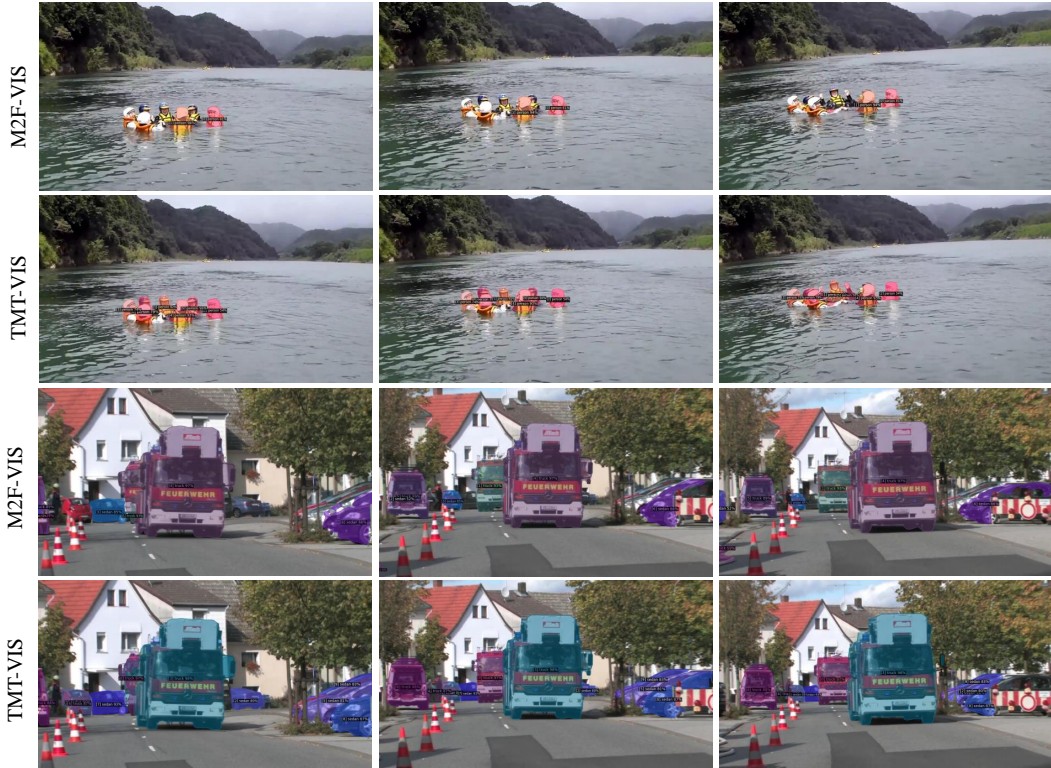

Figure 3: Visual comparison of our model with Mask2Former-VIS (abbreviated as 'M2F-VIS'). Our TMT-VIS shows better precision in segmenting small instances, e.g., the swimmers in the middle of the image, and classifying similar instances, e.g., truck and sedan are similar categories, their appearances are similar but different in sizes.

## 4.5 Visualization

We also conduct qualitative evaluations and the visual comparison with Mask2Former-VIS. Results are illustrated in Fig. 4. Our TMT-VIS demonstrates better capacity in segmenting small instances and classifying instances with similar categories. The reason is that our TMT-VIS model is benefited from our taxonomy compilation module with multi-dataset training.

## 5   Conclusion

In this paper, we propose TMT-VIS to address the dilemma in multiple dataset joint training in VIS. By developing a two-stage module, Taxonomy Compilation Module (TCM) and Taxonomy Injection Module (TIM), our new algorithm is able to train and utilize multiple datasets effectively by incorporating taxonomy guidance into the DETR-based model. Our proposed TMT-VIS harvests great performance improvements over the baselines and sets new state-of-the-art records on multiple popular and challenging VIS datasets and benchmarks. We hope that our new approach can provide valuable insights and motivate future VIS research.

**Limitations.** The utilization and aggregation of taxonomic embeddings may need further investigations, and there might be more sophisticated ways of associating taxonomic embeddings with multi-level queries that may further improve the performance.

**Broader impacts.** TMT-VIS is designed for effectively training multiple VIS datasets which achieves promising performance. We hope that TMT-VIS can have a positive impact on many industrial areas where dataset biases are severe. We would like to note that VIS research should mind not violate personal privacy.

**Acknowledgement.** This work is partially supported by the National Natural Science Foundation of China (No. 62201484), National Key R&D Program of China (No. 2022ZD0160100), HKU Startup Fund, and HKU Seed Fund for Basic Research.

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

# Appendix

This appendix provides more details about the proposed TMT-VIS, further details of VIS datasets, more qualitative visual comparisons.

## A    Dataset Details

Here, we provide a detailed overview of various VIS datasets in Table 8. Our extensive experimental evaluations are conducted on four challenging benchmarks, namely YouTube-VIS 2019 and 2021 [47], OVIS [35], and UVO [39]. YouTube-VIS 2019 [47] was the first large-scale dataset designed for video instance segmentation, comprising 2.9K videos averaging 4.61s in duration and 27.4 frames in validation videos. YouTube-VIS 2021 [47] poses a greater challenge with longer and more complex trajectory videos, averaging 39.7 frames in validation videos. The OVIS [35] dataset is another challenging VIS dataset with 25 object categories, focusing on complex scenes with significant object occlusions. Despite containing only 607 training videos, OVIS's videos last an average of 12.77s. Lastly, the UVO [39] dataset consists of 1.2K Kinetics-400 [20] videos, densely annotated at 30fps, featuring 81 object categories, including an extra "other" category for non-COCO instances. It provides exhaustive segmentation masks for all object instances in the 503 videos.

Among all categories in Youtube-VIS 2019, OVIS, and UVO, there are overlapping categories between each dataset, and there are also different categories that share similar semantics. The detailed overlapping categories are marked in Table 9. Overall, Youtube-VIS 2021 and OVIS share a more similar taxonomy space with Youtube-VIS 2019 than UVO, with a common category set of 34 out of 40 for Youtube-VIS 2021 and 22 out of 25 for OVIS. Typically, when the taxonomy spaces of datasets are similar, training them jointly will have smaller dataset biases, which leads to a better result in performance. The characteristics of these datasets align with the improvement in performance when validating the joint-training models on various datasets: the increase is more significant on Youtube-VIS 2021 and OVIS than on UVO. Further details of some specific categories can be found in Table 11.

Table 8: Key statistics of popular VIS datasets. Note that in UVO, the majority of the videos are for Video Object Segmentation, and only 503 videos are annotated for the VIS task. 'YTVIS' is the acronym of 'Youtube-VIS'.

|                   | YTVIS19 | YTVIS21 | OVIS | UVO    |
|-------------------|---------|---------|------|--------|
| Videos            | 2883    | 3859    | 901  | 11228  |
| Categories        | 40      | 40      | 25   | 81     |
| Instances         | 4883    | 8171    | 5223 | 104898 |
| Masks             | 131K    | 232K    | 296K | 593K   |
| Masks per Frame   | 1.7     | 2.0     | 4.7  | 12.3   |
| Object per Video  | 1.6     | 2.1     | 5.8  | 9.3    |

## B    Additional Ablation Studies

In this section, we provide more experiments on our proposed methods, we will discuss the generality and zero-shot properties of our training approach, and we will provide further details of the performance change in different taxonomies.

**Generality property.**    The proposed new taxonomy-aware training strategy is an effective and general strategy that can be adapted into various DETR-based approaches (both online & offline) and into various datasets. As in Table 10, when adding our TCM&TIM strategy to the popular VIS architecture Mask2Former-VIS [8], VITA [16], and IDOL [45], we harvest performance gains on all of the three challenging benchmarks. In OVIS, the increase can be up to 6.3% AP for Mask2Former-VIS. As for the popular IDOL, our strategy can also bring about an increase of 2.8% AP in performance. This demonstrates that the proposed taxonomy-aware module can be treated as a plug-and-play design that can be used in various DETR-based VIS methods (both online & offline) across different scenarios and all popular VIS benchmarks.

**Per-category performance.** In Table 11, we present the comparison in performance between Mask2Former-VIS and our TMT-VIS on several specific taxonomies. When datasets other than YTVIS have no such taxonomy, training Mask2Former-VIS on multiple datasets will end up decreasing the performance of such category, as shown in 'Duck' case from Table 11. When applying our approach, we can obtain a performance improvement due to that taxonomy information of 'Duck' is compiled and injected to the instance queries. In other taxonomies, such as 'Person' which appears across all VIS datasets, the improvements are also significant.

**Zero-shot property.** Further, our TMT-VIS can also perform well on zero-shot learning. We conducted the experiments on Youtube-VIS 2019 [47], OVIS [35], and UVO [39] benchmarks. As exhibited in Table 12, TMT-VIS can be utilized to transfer the knowledge of other VIS datasets to another dataset with a significant increase of 4.5% AP and 3.8% AP respectively. The extra taxonomy information provided by our newly designed TCM & TIM improve the model's performance when dealing with unfamiliar taxonomies.

## C  Visualization

In the visualization comparisons between Mask2Former-VIS and our model, we select some cases under different scenarios, which include setting with multiple similar instances, setting with fast-moving objects, and setting with different poses of instance.

In Fig. 4, we demonstrate videos with multiple similar instances, and TMT-VIS can both segment and track them more accurately than Mask2Former-VIS. The swimmers or the cyclists are all instances that belong to 'person' category, and TMT-VIS shows better segmentation and tracking. In Fig. 5, we present example videos which have quick movements in camera's perspective and have instances with different poses. In the top two rows, the sedan and the truck have similar appearances, and our model can distinguish and segment them with higher confidence (90% over 70%). In the last two rows, our model successfully segments the person in different poses, while Mask2Former-VIS fails to segment this person's arm in the first frame. However, the first two rows of Fig. 5 also show that our model still have the problem of segmenting instances with heavy occlusions. This suggests that simply combining taxonomy information is insufficient of solving severely occluded scenes, and that more information should be aggregated to instance queries to make the model more robust in segmenting instances.

Table 9: Overlapping categories of multiple VIS datasets with Youtube-VIS 2019 dataset.'YTVIS' is the acronym of 'Youtube-VIS'. As demonstrated in the table, YTVIS2021 and OVIS have a more similar taxonomy space, with 34 and 22 overlapping categories respectively.

| YTVIS19_40_categories | YTVIS21_40_categories | OVIS_25_categories | UVO_81_categories |
|---|---|---|---|
| Overlapping Categories | 34 | 22 | 19 |
| Person | ✓ | ✓ | ✓ |
| Giant_panda | ✓ | ✓ | |
| Lizard | ✓ | ✓ | |
| Parrot | ✓ | ✓ | |
| Skateboard | ✓ | | ✓ |
| Sedan | | ✓ | ✓ |
| Ape | | | |
| Dog | ✓ | ✓ | ✓ |
| Snake | ✓ | | |
| Monkey | ✓ | ✓ | |
| Hand | | | |
| Rabbit | ✓ | ✓ | |
| Duck | ✓ | | |
| Cat | ✓ | ✓ | ✓ |
| Cow | ✓ | ✓ | ✓ |
| Fish | ✓ | ✓ | |
| Train | ✓ | | ✓ |
| Horse | ✓ | ✓ | ✓ |
| Turtle | ✓ | ✓ | |
| Bear | ✓ | ✓ | ✓ |
| Motorbike | ✓ | ✓ | ✓ |
| Giraffe | ✓ | ✓ | ✓ |
| Leopard | ✓ | | |
| Fox | ✓ | | |
| Deer | ✓ | | |
| Owl | | | |
| Surfboard | | | ✓ |
| Airplane | ✓ | ✓ | ✓ |
| Truck | ✓ | ✓ | ✓ |
| Zebra | ✓ | ✓ | ✓ |
| Tiger | ✓ | ✓ | |
| Elephant | ✓ | ✓ | ✓ |
| Snowboard | ✓ | | ✓ |
| Boat | ✓ | ✓ | ✓ |
| Shark | ✓ | | |
| Mouse | ✓ | | |
| Frog | ✓ | | |
| Eagle | | | |
| Earless_seal | ✓ | | |
| Tennis_racket | ✓ | | ✓ |

Table 10: Ablation study on the generality property of TCM/TIM design with ResNet-50 backbone on multiple datasets.

| Datasets | Method | AP | $AP_{50}$ | $AP_{75}$ |
|---|---|---|---|---|
| YouTube-VIS 2019 | Mask2Former-VIS | 46.4 | 68.0 | 50.0 |
| | + TCM/TIM | 49.7 (↑ 3.3) | 73.4 (↑ 5.4) | 53.9 (↑ 3.9) |
| | VITA | 49.8 | 72.6 | 54.5 |
| | + TCM/TIM | 52.6 (↑ 2.8) | 74.4 (↑ 1.8) | 57.6 (↑ 3.1) |
| | IDOL | 49.5 | 74.0 | 52.9 |
| | + TCM/TIM | 51.4 (↑ 1.9) | 74.9 (↑ 0.9) | 55.0 (↑ 2.1) |
| YouTube-VIS 2021 | Mask2Former-VIS | 40.6 | 60.9 | 41.8 |
| | + TCM/TIM | 44.9 (↑ 4.3) | 66.1 (↑ 5.2) | 48.5 (↑ 6.7) |
| | VITA | 45.7 | 67.4 | 49.5 |
| | + TCM/TIM | 48.3 (↑ 2.6) | 69.8 (↑ 2.4) | 50.8 (↑ 1.3) |
| | IDOL | 43.9 | 68.0 | 49.6 |
| | + TCM/TIM | 45.8 (↑ 1.9) | 69.2 (↑ 1.2) | 50.9 (↑ 1.3) |
| OVIS | Mask2Former-VIS | 16.5 | 36.5 | 14.6 |
| | + TCM/TIM | 22.8 (↑ 6.3) | 43.6 (↑ 7.1) | 21.7 (↑ 7.1) |
| | VITA | 19.6 | 41.2 | 17.4 |
| | + TCM/TIM | 25.1 (↑ 5.5) | 45.9 (↑ 4.7) | 23.8 (↑ 6.4) |
| | IDOL | 30.2 | 51.3 | 30.0 |
| | + TCM/TIM | 33.0 (↑ 2.8) | 55.7 (↑ 4.4) | 33.2 (↑ 3.2) |

Table 11: Comparisons between per-category performance of Mask2Former-VIS and TMT-VIS. 'MDT' refers to 'Multiple Datasets Training', indicating whether the approach is trained on YTVIS, OVIS, and UVO. 'In Corresponding Dataset' is used to demonstrate whether the category is contained in corresponding dataset.

| Categories | Methods | In Corresponding Dataset | | | MDT | Test Set | AP |
|---|---|---|---|---|---|---|---|
| | | YTVIS | OVIS | UVO | | | |
| Person | Mask2Former-VIS | ✓ | | | | YTVIS | 57.2 |
| | TMT-VIS | ✓ | | | | YTVIS | 57.9 (↑ 0.7) |
| | Mask2Former-VIS | ✓ | ✓ | ✓ | ✓ | YTVIS | 59.3 |
| | TMT-VIS | ✓ | ✓ | ✓ | ✓ | YTVIS | 60.7 (↑ 1.4) |
| Duck | Mask2Former-VIS | ✓ | | | | YTVIS | 41.6 |
| | TMT-VIS | ✓ | | | | YTVIS | 42.4 (↑ 0.8) |
| | Mask2Former-VIS | ✓ | | | ✓ | YTVIS | 38.3 |
| | TMT-VIS | ✓ | | | ✓ | YTVIS | 43.9 (↑ 5.6) |
| Monkey | Mask2Former-VIS | ✓ | | | | YTVIS | 24.7 |
| | TMT-VIS | ✓ | | | | YTVIS | 26.4 (↑ 1.7) |
| | Mask2Former-VIS | ✓ | ✓ | | ✓ | YTVIS | 25.6 |
| | TMT-VIS | ✓ | ✓ | | ✓ | YTVIS | 29.1 (↑ 3.5) |
| Snowboard | Mask2Former-VIS | ✓ | | | | YTVIS | 8.9 |
| | TMT-VIS | ✓ | | | | YTVIS | 11.8 (↑ 2.9) |
| | Mask2Former-VIS | ✓ | | ✓ | ✓ | YTVIS | 10.0 |
| | TMT-VIS | ✓ | | ✓ | ✓ | YTVIS | 14.5 (↑ 4.5) |

Table 12: Zero-shot Performance of TMT-VIS with ResNet-50 backbone. The results demonstrate the zero-shot ability of our proposed method. YouTube-VIS 2019 is abbreviated as 'YTVIS'.

| Method | Train Set | | | Test Set | AP | AP$_{50}$ | AP$_{75}$ |
| | YTVIS | OVIS | UVO | | | | |
| --- | --- | --- | --- | --- | --- | --- | --- |
| Mask2Former-VIS | | ✓ | ✓ | YTVIS | 7.1 | 11.4 | 8.3 |
| TMT-VIS | | ✓ | ✓ | YTVIS | 11.6 (↑ 4.5) | 17.2 (↑ 5.8) | 15.0 (↑ 6.7) |
| Mask2Former-VIS | ✓ | | ✓ | OVIS | 3.7 | 9.8 | 5.2 |
| TMT-VIS | ✓ | | ✓ | OVIS | 7.5 (↑ 3.8) | 14.1 (↑ 4.3) | 8.5 (↑ 3.3) |

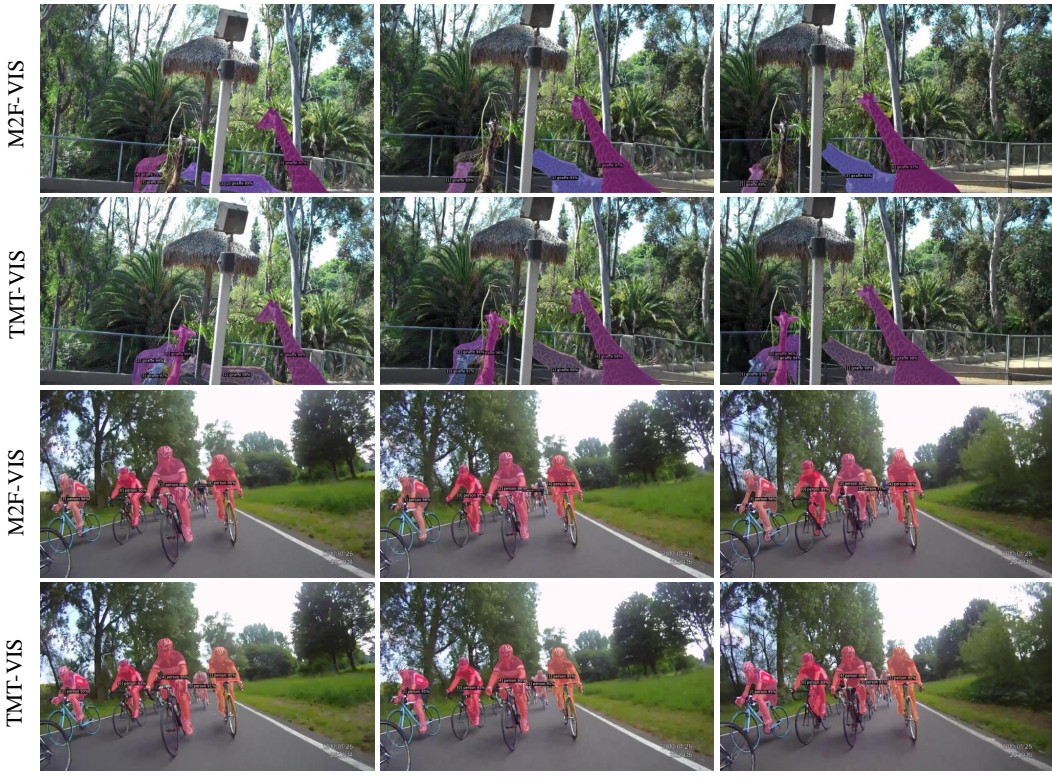

Figure 4: Visual comparison of our model with Mask2Former-VIS (abbreviated as 'M2F-VIS'). Our TMT-VIS shows better precision in segmenting and tracking small instances with the same taxonomy, such as the giraffes from the first two rows or the cyclists in the last two rows, and TMT-VIS shows better performance).

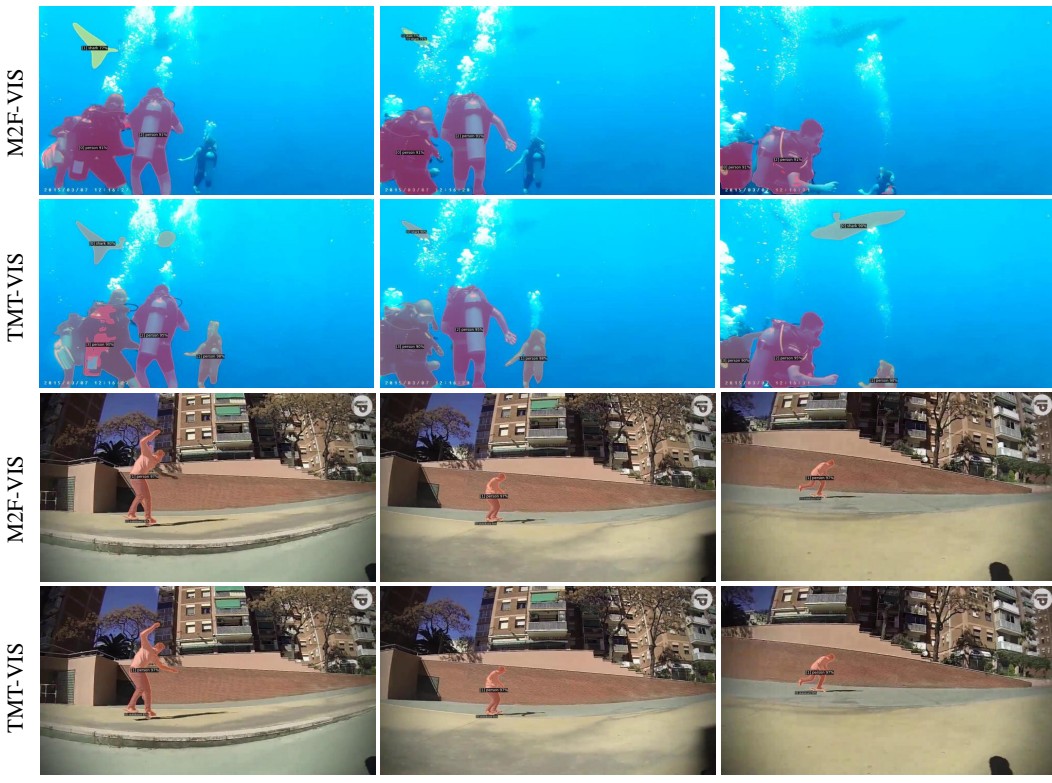

Figure 5: Visual comparison of our model with Mask2Former-VIS (abbreviated as 'M2F-VIS'). Our TMT-VIS shows better precision in segmenting and tracking instances with occlusions. In the top two rows, TMT-VIS could successfully segment the shark hidden behind the bubbles as well as the diver in the middle, while M2F-VIS fails to segment these instances. In the last two rows, our model successfully segments the person in different poses, while M2F-VIS fails to segment this person's arm in the first frame.

