# OpenReview forum: "TMT-VIS: Taxonomy-aware Multi-dataset Joint Training for Video Instance Segmentation"
_NeurIPS.cc/2023/Conference — NeurIPS 2023 poster_

### Official Review · Reviewer_k3N8 · 2023-07-05

**Soundness:** 3 good
**Presentation:** 3 good
**Contribution:** 2 fair
**Rating:** 5
**Confidence:** 4

**Summary:**

The paper proposes TMT-VIS, combining multiple VIS datasets for a unified VIS model. Specifically, TMT-VIS introduces taxonomy embedding as prompts to make the model aware of taxonomy from different datasets. Experiments on YVIS-2019, YVIS-2021, OVIS, and UVO demonstrate the effectiveness of dataset unification.


**Strengths:**

+ The proposed method is straightforward and easy to follow. The performance under joint training is good.
+ Unifying multiple datasets into a single model is of great value.


**Weaknesses:**

- The authors claim that "TMT-VIS is the first DETR-style framework that can jointly train multiple video instance segmentation datasets with such a huge improvement." However, previous work UNINEXT [1] also train multiple VIS dataset and achieves a performance gain with a DETR-style framework, which is missed in the related work.
- Tab.3 is confusing. The authors are recommended to evaluate all three datasets in this table to show the performance gain under joint training. Furthermore, the authors claim that Mask2Former-VIS meets difficulties in dealing with datasets, decreasing performance under joint training. Does it indicate the ID-I and ID-V in Tab.3? It is worth noticing that TMT-VIS jointly trained on YTVIS and UVO also decreases compared to the model trained on a single dataset. In other words, the proposed TMT-VIS does not overcome performance drop under joint training. It is ambiguous whether the performance gain is caused by the proposed method boosting the performance on a single dataset or alleviating the conflict in multi-dataset training.
- The ablation on the size of taxonomic embedding is not sufficient. For example, what's the performance using all embeddings? Does it have negative effects when the topk taxonomic embeddings miss the gt classes?
- Please provide more details of the taxonomy-aware matching loss.

[1] Universal Instance Perception as Object Discovery and Retrieval

**Questions:**

please see my listed questions in the weakness part.


**Limitations:**

The authors have adequately addressed their work's limitations and potential negative societal impact.

---

> ### Author Rebuttal · Authors · 2023-08-09
>
> At the beginning, we want to thank you for the detailed, insightful and constructive comments.
>
> ### UNINEXT
> Thanks for pointing out our wrong claim. We acknowledge that the UNINEXT is the first DETR-based method which jointly trains multiple VIS datasets. It simply utilizes BERT language encoder to generate language embeddings of categories from all video datasets, and fuse the information with visual embeddings through a simple bi-directional cross-attention module.
>
> However, we argue that we are the current SOTA method in jointly training multiple VIS datasets. Our method, on the other hand, adjusts the language embeddings via a taxonomy compilation module which consists of a spatio-temporal adapter and a FFN network, as well as of a taxonomy injection module which fuses the taxonomy information into video queries with a multi-level cross-attention and self-attention layers. Moreover, it achieves a lower performance of 64.3 AP (ours is 64.9 AP) with more training data on Youtube-VIS 2019.
>
> Table 5-1. Comparison with UNINEXT on YTVIS-19.
> | Method        | Datasets Number | Training time  | Backbone   | AP   |
> | ------------- | :-------------: | -------------- | ---------- | ---- |
> | UNINEXT       | 8               | 3 days         | ConvNext-L | 64.3 |
> | TMT-VIS(Ours) | 4               | 1 day 12 hours | Swin-L     | 64.9 |
>
> ### Ablation study on the size of taxonomic embedding
>
> Thanks for your careful suggestions about the size of taxonomic embedding. We conducted more ablation studies on this based on the ResNet-50 backbone, and the experiment is tested on YTVIS-19 dataset. As the results shown in Table 5-2, the size of taxonomic embedding is crucial to the performance. When the size is set to 1 or 2, which is apparently less than the overall categories in input video, topk taxonomic embeddings miss the gt classes. Compilation of such taxonomic information and injecting them into video queries provide no guidance to the convergence but diverge the attention of queries to irrelevant categories, and thus results in no improvement and even degradation in the final results. When using all embeddings, the improvement is trivial since TCM can not provide filtration to irrelevant classes, and the TIM module is simply injecting the information of the whole category space to queries.
>
> Table 5-2. Ablation study on sizes of taxonomic embedding set in TCM.
>
> | Size | AP   | Size | AP   |
> | ---- | :--: | ---- | :--: |
> | 1    | 47.1 | 15   | 49.2 |
> | 2    | 47.5 | 20   | 48.5 |
> | 5    | 49.3 | 50   | 47.4 |
> | 10   | 49.7 | 100  | 47.2 |
>
> ### The reorganization of Tab.3
>
> Thanks for your advice on our table. In this part, we update these experiments and reorganize Tab.3, and the results are shown in the rebuttal pdf.
>
> We present Tab.3 to demonstrate that due to the heterogeneity of category spaces of multiple VIS datasets, simply training on multiple datasets with an aggregated category space will not improve the performance significantly. The performance may even drop due to the huge imbalance of data, and that ID-I and ID-V in Tab.3 is an example in showing this phenomenon.
>
> ### Performance degradation when joint training UVO and YTVIS
>
> YTVIS-19 takes advantage of an existing dataset called YouTube-VOS. OVIS is collected specifically for video instance segmentation in occluded scenes. UVO, on the other hand, adopts videos from Kinetics-400, which are human-centric and contain diverse sets of human actions and human-object interactions, and UVO is densely annotated with many more instances. The key statistics of these datasets are shown in the rebuttal pdf, which corresponds to the Tab.1 of the supplementary file.  As illustrated in the table, the imbalance between these VIS datasets is significant. When jointly training YTVIS-19 with UVO, the model tends to converge to fit UVO, which leads to the degradation of performance on validation on YTVIS-19. When jointly training YTVIS-19 and UVO with OVIS, the imbalance is alleviated, and results in a notable performance improvement. It’s worth noting that due to the scale of VIS datasets, especially YTVIS-19, the final results have a fluctuation of approximately 0.3AP, and this may also be the direct reason for the seemingly drop of our TMT-VIS compared with Mask2Former-VIS because in other training settings our TMT-VIS significantly outperforms Mask2Former-VIS.
>
> ### More details of the taxonomy-aware matching loss
>
> The formula of the taxonomy-aware matching loss is shown as follows:
>
> $$
> \mathcal{L} = \sum\mathcal{L}^{\text{ce}}(\mathbf{P^c}, \mathbf{G^c}) + \sum\mathcal{L}_{inj}^{\text{ce}}(\mathbf{P^{c^{\prime}}},\mathbf{G^{c^{\prime}}}) + \sum \mathcal{L}^{\text{dice}}(\mathbf{P^m}, \mathbf{G^m}) + \sum \mathcal{L}^{\text{bce}}(\mathbf{P^m}, \mathbf{G^m})
> $$
>
> where $\mathcal{L}^{\text{ce}}$ denotes the cross-entropy loss for classification, $\mathcal{L}_{inj}^{\text{ce}}$ denotes the extra `Taxonomy-aware matching loss',which is also cross-entropy loss, except for the predictions and gts being different. The
> $\mathcal{L}^{\text{dice}}$,$\mathcal{L}^{\text{bce}}$ denotes the dice loss and binary cross-entropy loss between mask predictions and matched ground truths, which are the same as the strategy in Mask2Former, where we sample different sets of $K=12544=122 \times 122$ points for different pairs of prediction and ground truth using importance sampling. Here $\mathbf{P}$ is the prediction, and $\mathbf{G}$ is the ground truth, $\mathbf{c}$ refer to classification while $\mathbf{m}$ refer to mask, and $\mathbf{c^{\prime}}$ refers to classification right after the injection process. $\mathbf{G^{c^{\prime}}}$ refers to ground truth categories to the taxonomy represented by the predicted set of taxonomic embeddings. By adding this supervision, we further guarantee the guidance provided by the compiled taxonomic embeddings.

---

> > ### Comment · Reviewer_k3N8 · 2023-08-17
> >
> > I would like to thank the authors for the detailed rebuttal. Most of my concerns are addressed. The proposed method shows non-trivial performance in unifying multiple video segmentation datasets into a single model. However, I agree with Revier 5w4z's concern about the uniqueness of video tasks. For instance, UNINEXT unifies not only VOS but also other detection, segmentation, and reference comprehension tasks on both video and image. It is somehow unclear the motivation for how the proposed method target boosting VOS instead of image segmentation, or the authors just follow the trend of multi-dataset training. I expect the authors to provide deep insights into how the proposed method effectively unifies the multi-dataset VOS and boosts the performance of each dataset.

---

> > > ### Author Response · Authors · 2023-08-20
> > >
> > > We extend our deepest appreciation for the invaluable time and selfless efforts you have dedicated to reviewing and providing insightful comments on our paper.
> > >
> > > In fact, our research focus remains on VIS task rather than image instance segmentation, and we are open for the results (no matter improving or degrading) on jointly training image datasets. Our research motivation is that current image instance segmentation datasets are significantly larger than VIS datasets. LVIS, for example, has 160k images and 2M instance mask annotations. The scenarios contained in these images are thus more varied than VIS datasets. Such image dataset is so large and dominant that jointly training multiple image instance seg datasets become less exciting and influential.
> > >
> > > On the other hand, VIS datasets are smaller in scale, and what we possess are numerous isolated field-specific datasets, rather than dominant scale datasets. Due to this fact, combining these varied VIS datasets and training them jointly become useful and meaningful for VIS research. Also, in our early experiments we noticed that adapting pretrained MAE-based weights (ImageMAE \& VideoMAE ) to Mask2Former structure is unsuccessful (please refer to Table 1). Based upon these, we are motivated to research on applying joint training methods in VIS training, hoping to unify these separate specific datasets.
> > >
> > > A major challenge in multiple-dataset joint training is the heterogeneity of multiple datasets’ category space. As a result of the difference in category space, though mask precision increases with the data volume, dataset biases might hinder models from generalization: simply utilizing multiple datasets will dilute the attention of models on different categories. Therefore, increasing the data scale and enriching label space while improving classification precision become a huge challenge for researchers. This phenomenon was shown in our ablation studies.
> > >
> > > Our method, adjusts the language embeddings via a taxonomy compilation module which consists of a spatio-temporal adapter and a FFN network, as well as of a taxonomy injection module which fuses the taxonomy information into video queries with a multi-level cross-attention and self-attention layers. In the taxonomy compilation module, taxonomy embeddings are designed to interact with video features in order to unearth the potential taxonomy contained in each of the video frames. After this, we calculated the dot product between the different modulated taxonomic embeddings to predict the most relevant taxonomy in the given video.
> > >
> > > By compiling and injecting the possible taxonomic information to queries as guidance, queries tend to converge to the desired instances and desired categories faster and finally come up with better precision. Also, when we constrain the most relevant taxonomy of input video, we no longer have to worry about the full heterogeneous category space.
> > >
> > > As depicted in Table 2, even when training categories that only exist in one VIS dataset, our method still manages to enhance its Average Precision (AP).
> > >
> > > Table 1. Experiments on Adapting ImageMAE \& VideoMAE to VIS tasks
> > > | Method       | Backbone | AP   |
> > > | ------------ | -------- | ---- |
> > > | M2F          | SWIN-B   | 59.5 |
> > > | ImageMAE+M2F | VIT-B    | 53.3 |
> > > | VideoMAE+M2F | VIT-B    | 27.1 |
> > >
> > > Table 2. Comparisons between per-category performance of Mask2Former-VIS and TMT-VIS. ‘MDT’ refers to ‘Multiple Datasets Training’, indicating whether the approach is trained on YTVIS, OVIS,
> > > and UVO. The '√' in the middle column is used to demonstrate whether the category is contained in
> > > corresponding dataset. For example, all datasets have the 'person' category, and thus all datasets have the corresponding '√'.
> > > | Categories | Methods | YTVIS | OVIS | UVO  | MDT  | AP   |
> > > | ---------- | ------- | :---: | :--: | :--: | :--: | :--: |
> > > | Person     | M2F     | √     |      |      |      | 57.2 |
> > > |            | TMT-VIS | √     |      |      |      | 57.9 |
> > > |            | M2F     | √     | √    | √    | √    | 59.3 |
> > > |            | TMT-VIS | √     | √    | √    | √    | 60.7 |
> > > | Duck       | M2F     | √     |      |      |      | 41.6 |
> > > |            | TMT-VIS | √     |      |      |      | 42.4 |
> > > |            | M2F     | √     |      |      | √    | 38.3 |
> > > |            | TMT-VIS | √     |      |      | √    | 43.9 |
> > > | Monkey     | M2F     | √     |      |      |      | 24.7 |
> > > |            | TMT-VIS | √     |      |      |      | 26.7 |
> > > |            | M2F     | √     | √    |      | √    | 25.6 |
> > > |            | TMT-VIS | √     | √    |      | √    | 29.1 |
> > > | Snowboard  | M2F     | √     |      |      |      | 8.9  |
> > > |            | TMT-VIS | √     |      |      |      | 11.8 |
> > > |            | M2F     | √     |      | √    | √    | 10.0 |
> > > |            | TMT-VIS | √     |      | √    | √    | 14.5 |

---

### Official Review · Reviewer_VJ6M · 2023-07-06

**Soundness:** 2 fair
**Presentation:** 3 good
**Contribution:** 2 fair
**Rating:** 4
**Confidence:** 5

**Summary:**

Due to the lack of large-scale datasets in the VIS task, the authors propose a multi-dataset joint training method. Due to the heterogeneity of the category spaces of different datasets, simply stacking datasets may lead to performance degradation. For this situation, the author designs a two-stage classification aggregation module, which first compiles the classification information of videos from different datasets, and then Aggregating these categorical information translates categorical priors into instance queries for better performance. The two-stage modules include a classification compilation module and a classification injection module. The former compiles classification information, and the latter utilizes TCM classification information to inject guidance for queries. Experiments prove the effectiveness of this method.

**Strengths:**

The main advantage of this paper is that the proposed joint training model for multiple datasets, which uses taxonomy information to mitigate the heterogeneity in the category space of different datasets, has achieved good results and can try to migrate this approach to other fields.

**Weaknesses:**

The paper lacks clear explanations on the description of the module and the loss function.
In the experimental comparison, the performance in the ovis dataset is not satisfactory.
Furthermore, the generalization ability of this work on different datasets is limited.

**Questions:**

1.In Taxonomy Compilation Module, what is the specific form or dimension of label space like? Is it a set of category names?
2. Taxonomy-aware matching loss, mask loss, and cls loss - what are the specific formulas for these three losses?
3.In Table 2, Can the effectiveness of the proposed method be verified as it has lower performance on OVIS dataset compared to both IDOL and GenVIS methods?
4.In Experiment V in Table 3, when jointly training with the Youtube-VIS 2019 and UVO datasets, the Mask2Former-VIS method exhibits a decrease of 0.2 in AP compared to using only YTVIS, but it shows an improvement of 3.2 in AP50. However, the TMT-VIS method experiences a decrease of 0.3 in AP and 0.5 in AP50. Can we conclude that the model lacks robustness to differences between different datasets?

**Limitations:**

Please see the weakness.

---

> ### Author Rebuttal · Authors · 2023-08-09
>
> At the beginning, we want to thank you for the detailed, insightful and constructive comments.
>
> ### Label space in Taxonomy Compilation Module
> The specific dimensions of label space are $K \times D$, where $K$ refers to the total number of categories in datasets, and $D$ represents the hidden dimensions of outputs of text encoders.
>
> ### Formula of taxonomy-aware matching loss, mask loss, and classification loss
>
> The formula of different losses are shown below:
>
> $$
> \mathcal{L} = \sum\mathcal{L}^{\text{ce}}(\mathbf{P^c}, \mathbf{G^c}) + \sum\mathcal{L}_{inj}^{\text{ce}}(\mathbf{P^{c^{\prime}}},\mathbf{G^{c^{\prime}}}) + \sum \mathcal{L}^{\text{dice}}(\mathbf{P^m}, \mathbf{G^m}) + \sum \mathcal{L}^{\text{bce}}(\mathbf{P^m}, \mathbf{G^m})
> $$
>
> where $\mathcal{L}^{\text{ce}}$ denotes the cross-entropy loss for classification, $\mathcal{L}_{inj}^{\text{ce}}$ denotes the extra ‘Taxonomy-aware matching loss’, which is also cross-entropy loss, except for the predictions and gts being different. The  $\mathcal{L}^{\text{dice}}$,$\mathcal{L}^{\text{bce}}$ denotes the dice loss and binary cross-entropy loss between mask predictions and matched ground truths, which are the same as the strategy in Mask2Former, where we sample different sets of $K=12544=122 \times 122$ points for different pairs of prediction and ground truth using importance sampling. Here $\mathbf{P}$ is the prediction, and $\mathbf{G}$ is the ground truth, $\mathbf{c}$ refer to classification while $\mathbf{m}$ refer to mask, and $\mathbf{c^{\prime}}$ refers to classification right after the injection process. $\mathbf{G^{c^{\prime}}}$ refers to ground truth categories to the taxonomy represented by the predicted set of taxonomic embeddings. By adding this supervision, we further guarantee the guidance provided by the compiled taxonomic embeddings are successfully injected.
>
> ### The effectiveness of TMT-VIS on OVIS
>
> Our method is built upon Mask2Former and VITA, which were the SOTA offline method. Also, our module can be added to SOTA method Gen-VIS \& IDOL, which will also boost their performance significantly, as shown in Table.4-1. The results verify the effectiveness of our proposed method, which can be plug-and-play to both online and offline VIS solutions. Also, the parameters of our added design are not tuned, so that the performance could be even higher with further experiments.
>
> Table 4-1. Experiments on the effectiveness of TMT-VIS on OVIS.
>
> | Method | Multi-dataset? | w/o our design | Backbone  | AP   |
> | ------ | :--------------: | :------------: | --------- | :--: |
> | IDOL   |                   |                | ResNet-50 | 30.2 |
> | IDOL   | √                 |                | ResNet-50 | 32.1 |
> | IDOL   | √                 | √              | ResNet-50 | 33.6 |
> | GenVIS |                   |                | ResNet-50 | 35.8 |
> | GenVIS | √                 |                | ResNet-50 | 37.3 |
> | GenVIS | √                 | √              | ResNet-50 | 38.4 |
>
> ### Robustness of our method across various datasets
>
> It's true that there are degradation when jointly training UVO and YTVIS-19, but we argue that in most of the training settings TMT-VIS improves significantly better than Mask2Former (see the table in rebuttal pdf), and so this result can't indicate that our method lacks robustness. In the following we try to explain the reason why this may occur.
>
> The drop in performance may be caused by the imbalance in scale among VIS datasets. YTVIS-19 takes advantage of an existing dataset called YouTube-VOS.  OVIS is collected specifically for video instance segmentation in occluded scenes. UVO, on the other hand, adopts videos from Kinetics-400, which are human-centric and contain diverse sets of human actions and human-object interactions, and UVO is densely annotated with many more instances.  The key statistics of these datasets are shown in the following table, which corresponds to the Table.1 of the supplementary file.  As illustrated in the table, the imbalance between these VIS datasets is significant. When jointly training YTVIS-19 with UVO, the model tends to converge to fit UVO, which leads to the degradation of performance on validation on YTVIS-19. When jointly training YTVIS-19 and UVO with OVIS, the imbalance is alleviated, and results in a notable performance improvement.
>
> It’s worth noting that due to the scale of VIS datasets, especially YTVIS-19, the final results usually have a fluctuation of approximately 0.3AP, and this may also be the direct reason for the seemingly drop of our TMT-VIS compared with Mask2Former-VIS because in other training settings our TMT-VIS significantly outperforms Mask2Former-VIS.
>
> Table 4-2. Key statistics of multiple VIS datasets.
>
> |                  | YT19 | YT21 | OVIS | UVO    |
> | ---------------- | ---- | ---- | ---- | ------ |
> | Videos           | 2883 | 3859 | 901  | 11228  |
> | Categories       | 40   | 40   | 25   | 81     |
> | Instances        | 4883 | 8171 | 5223 | 104898 |
> | Masks            | 131k | 232k | 296k | 593k   |
> | Masks per Frame   | 1.7  | 2    | 4.7  | 12.3   |
> | Objects per Video | 1.6  | 2.1  | 5.8  | 9.3    |

---

> ### Comment · Reviewer_VJ6M · 2023-08-18
>
> The author made corresponding explanations and experiments on the robustness of the method. Whether the authors found performance fluctuations caused by dataset imbalance in the original work. In response to this problem, the author does not seem to solve the problem positively, but adopts another joint training strategy. Therefore，combining the comments of other reviewers and the author's responses, I still maintain the original score.

---

> > ### Author Response · Authors · 2023-08-20
> >
> > We deeply appreciate the valuable time and selfless efforts you have dedicated to reviewing and commenting our paper.
> >
> > Dataset imbalance can be roughly divided into two types: category imbalance and dataset scale imbalance. While it's true that scale imbalance could potentially impact the joint training of multiple datasets, we want to clarify that this isn't our primary research focus. In our methodology, we employ a widely used weighted sampling strategy to mitigate the imbalance across various VIS datasets, with ablation studies related to specific hyperparameters of this strategy detailed in Table 5 of our initially submitted paper.
> >
> > Our main research focus is on category imbalance (also referred to as class imbalance in some literature). We address this issue using a two-stage taxonomy aggregation module consisting of: the Taxonomy Correlation Module (TCM), designed to uncover potential taxonomy within each video frame by interacting taxonomic embeddings with video features; and the Taxonomy Integration Module (TIM), developed to aggregate modulated taxonomic embeddings that carry the most relevant taxonomy information. Some categories demonstrate marked improvements (as shown in our supplementary material). As depicted in Table 1, even when training categories that only exist in one VIS dataset, our method still manages to enhance its Average Precision (AP) despite significant category imbalance.
> >
> >
> > Table 1. Comparisons between per-category performance of Mask2Former-VIS and TMT-VIS. ‘MDT’ refers to ‘Multiple Datasets Training’, indicating whether the approach is trained on YTVIS, OVIS,
> > and UVO. The '√' in the middle column is used to demonstrate whether the category is contained in
> > corresponding dataset. For example, all datasets have the 'person' category, and thus all datasets have the corresponding '√'.
> > | Categories | Methods | YTVIS | OVIS | UVO  | MDT  | AP   |
> > | ---------- | ------- | :---: | :--: | :--: | :--: | :--: |
> > | Person     | M2F     | √     |      |      |      | 57.2 |
> > |            | TMT-VIS | √     |      |      |      | 57.9 |
> > |            | M2F     | √     | √    | √    | √    | 59.3 |
> > |            | TMT-VIS | √     | √    | √    | √    | 60.7 |
> > | Duck       | M2F     | √     |      |      |      | 41.6 |
> > |            | TMT-VIS | √     |      |      |      | 42.4 |
> > |            | M2F     | √     |      |      | √    | 38.3 |
> > |            | TMT-VIS | √     |      |      | √    | 43.9 |
> > | Monkey     | M2F     | √     |      |      |      | 24.7 |
> > |            | TMT-VIS | √     |      |      |      | 26.7 |
> > |            | M2F     | √     | √    |      | √    | 25.6 |
> > |            | TMT-VIS | √     | √    |      | √    | 29.1 |
> > | Snowboard  | M2F     | √     |      |      |      | 8.9  |
> > |            | TMT-VIS | √     |      |      |      | 11.8 |
> > |            | M2F     | √     |      | √    | √    | 10.0 |
> > |            | TMT-VIS | √     |      | √    | √    | 14.5 |

---

> > ### Comment · Area_Chair_kcB3 · 2023-08-20
> >
> > Please expand on why if 3/4 other reviews were positive why you still strongly believe the paper should be rejected? What reasons for acceptance do you disagree with from the other positive reviewers?

---

> > ### Author Response · Authors · 2023-08-21
> >
> > Dear Reviewer VJ6M,
> >
> > We express our heartfelt appreciation for the valuable time and selfless dedication you have invested in reviewing our paper.
> >
> > As the deadline for discussion draws near, we would like to inquire if our response adequately addressed your concerns. We are eager to engage in further discussion.
> >
> > Best regards,
> > Authors of paper 11160

---

### Official Review · Reviewer_fxcA · 2023-07-07

**Soundness:** 3 good
**Presentation:** 3 good
**Contribution:** 2 fair
**Rating:** 6
**Confidence:** 4

**Summary:**

This paper works on multi-dataset training on video instance segmentation. The authors built on top of Mask2Former, and introduced two components to enable the model to work under different label sets, and take advantages of the given label sets. The two components are both ablated in experiments. The overall framework improved Mask2Former on three popular datasets and achieved state-of-the-art performance.

**Strengths:**

- The problem on training a video instance segmentation model on multiple datasets is important. This paper takes a good step in this direction. The overall framework makes sense to me.

- The results are strong. The improvements over the Mask2Former and VITA baselines are significant, and are consistent under three datasets and two backbones.

- The contributions are well ablated in Table 4 - 7.

- The paper is well written and easy to follow.

**Weaknesses:**

- The most important Table, Table 3, is extremely hard to read. The numbers of different rows are on different datasets, making it hard to compare across rows. PLEASE split the columns by datasets. Please report numbers on all datasets, which should be feasible as the model used a CLIP classifier. We can remove AP50/ AP75 if space is a constraint.

- Please clarify: does the model require a known vocabulary during inference? Can it test on in-the-wild images (e.g., using a unified vocabulary)?

- [Optional] One other interesting aspect of multi-dataset training in VIS is the difference between the framerate/ image size across different datasets. If will be great if the authors can provide some discussion on this (if this is not trivial).

**Questions:**

Overall the paper works on an important task, proposes a valid model, and has great results. My concerns are mostly on presentation and clarity (see weekness). Please clarify them in the rebuttal.

**Limitations:**

Yes.

---

> ### Author Rebuttal · Authors · 2023-08-10
>
> At the beginning, we want to thank you for the detailed, insightful and constructive comments.
> ### Reorganization of Table 3
>
> In Table 5 of supplementary material, we have posted several experiments of zero-shot performance of our methods when compared with previous methods. In this part, we update these experiments and reorganize Table 3 in the submission file. The updated version is shown below:
>
> Table 3-1. Ablation study on training with multiple VIS datasets.
>
> |  ID  | Method  | YTVIS$_{train}$ | OVIS$_{train}$ | UVO$_{train}$  | YTVIS$_{val}$ | OVIS$_{val}$ | UVO$_{val}$  |
> | :--: | :-----: | :---: | :--: | :--: | :---: | :--: | :--: |
> |  1   |   M2F   |   √   |      |      | 46.4  | 2.3  | 1.9  |
> |  2   |   M2F      |       |  √   |      |  5.2  | 16.5 | 3.6  |
> |  3   |   M2F      |       |      |  √   |  4.4  | 2.5  | 18.2 |
> |  4   |   M2F      |   √   |  √   |      | 47.3  | 17.4 | 4.9  |
> |  5   |   M2F      |   √   |      |  √   | 46.2  | 3.7  | 19.0 |
> |  6   |   M2F      |       |  √   |  √   |  7.1  | 16.6 | 18.7 |
> |  7   |   M2F      |   √   |  √   |  √   | 47.2  | 17.2 | 19.3 |
> |  8   | TMT-VIS |   √   |      |      | 47.3  | 7.2  | 6.5  |
> |  9   | TMT-VIS        |       |  √   |      | 10.5  | 17.8 | 8.0  |
> |  10  |  TMT-VIS       |       |      |  √   | 10.1  | 8.6  | 18.8 |
> |  11  |  TMT-VIS      |   √   |  √   |      | 48.8  | 20.9 | 10.1 |
> |  12  |  TMT-VIS       |   √   |      |  √   | 47.0  | 10.3 | 20.4 |
> |  13  |  TMT-VIS       |       |  √   |  √   | 14.8  | 19.4 | 20.2 |
> |  14  |  TMT-VIS       |   √   |  √   |  √   | 49.7  | 22.8 | 21.2 |
>
>
> ### Test on in-the-wild image
> The tests on in-the-wild datasets can refer to the first answer. Unfortunately there are no other in-the-wild VIS datasets, and so we tested the performance on another in-the-wild dataset from a video-related task, which further proves the transferability of our design.
>
> We tested our method on VIPSeg[1] based on the popular Video K-Net[2], the results are shown in Table. VIPSeg is a large-scale dataset for video panoptic segmentation (VPS)[3], a task which aims to simultaneously predict object classes, bounding boxes, masks, instance id associations, and semantic segmentation in video frames. There are a total of 3,536 videos with 84,750 pixel-wise annotated frames in VIPSeg, which covers 232 scenarios with 124 categories, including 58 things’ classes and 66 stuff’s classes. Here, VPQ and STQ are both metrics from the VPS task. With our additional design, we compiled taxonomic information and injected them to thing and stuff kernels, and trained YTVIS and VIPSeg jointly, with the total training epochs set to 12. The results are shown below, with our design, the performance of Video K-Net increases with a 2.3% VPQ improvement and 2.6% STQ improvement. This further demonstrates the transferability of our design in in-the-wild scenarios.
>
> Table 3-2. Results on VIPSeg Dataset.
>
> | Method      | Multiple Dataset? | Backbone  | VPQ  | STQ  |
> | ----------- | :---------------: | --------- | ---- | ---- |
> | Video K-Net |                   | ResNet-50 | 26.1 | 33.1 |
> | Video K-Net | √                 | ResNet-50 | 27.6 | 34.8 |
> | TMT-VIS     | √                 | ResNet-50 | 28.4 | 35.7 |
>
> ### Vocabulary in inference
> In the inference part, our SOTA performance method utilizes the given vocabulary of the dataset. However, our method could still perform well with a unified vocabulary, which is credit to the utilization of CLIP encoder, just as Table 3-1 shows.
>
> ### Different frame rate across different datasets
>
> We resample the VIS datasets to a different frames per second(fps) when training on multiple datasets, and the results are shown as follows. Worth mentioning, all VIS datasets are annotated in a fps of 6 (UVO is 30) . As we can see, as the fps decrease from 6 to 1, the performance significantly drops. This is likely to be the consequence of learning temporal relations. With a sparse annotation, the displacement of objects between annotated frames become more significant, and thus increase the difficulty for queries to both segment the desired instances and track their trajectories.
>
> Table 3-3. Experiments on different dataset fps.
> | Method      | Testing Dataset | Dataset FPS | AP   |
> | ----------- | :-------------: | :---------: | :--: |
> | Mask2Former | YTVIS-19         | 1           | 45.5 |
> | TMT-VIS     | YTVIS-19         | 1           | 50.8 |
> | Mask2Former | YTVIS-19         | 6           | 46.4 |
> | TMT-VIS     | YTVIS-19         | 6           | 52.6 |
>
> ### Reference
>
> [1] Miao J, Wang X, Wu Y, et al. Large-scale video panoptic segmentation in the wild: A benchmark[C]//Proceedings of the IEEE/CVF Conference on Computer Vision and Pattern Recognition. 2022: 21033-21043.
>
> [2] Li X, Zhang W, Pang J, et al. Video k-net: A simple, strong, and unified baseline for video segmentation[C]//Proceedings of the IEEE/CVF Conference on Computer Vision and Pattern Recognition. 2022: 18847-18857.
>
> [3] Kim D, Woo S, Lee J Y, et al. Video panoptic segmentation[C]//Proceedings of the IEEE/CVF Conference on Computer Vision and Pattern Recognition. 2020: 9859-9868.

---

> > ### Comment · Reviewer_fxcA · 2023-08-18
> > **Thank you for the rebuttal**
> >
> > Thank you for the rebuttal. My concerns are all well addressed in the rebuttal. I keep my original positive rating.

---

### Official Review · Reviewer_wd7m · 2023-07-07

**Soundness:** 3 good
**Presentation:** 3 good
**Contribution:** 3 good
**Rating:** 5
**Confidence:** 4

**Summary:**

This article addresses video instance segmentation from a perspective of multi-dataset joint training. The proposed method is based on DETR, while the main contribution is to inject label taxonomy into model training based on CLIP. The model is evaluated on several standard benchmarks.

**Strengths:**

While multi-dataset training is not a new idea in such fields like object detection, the work is a pioneering study (as far as I know) to explore the idea in video instance segmentation. The paper is overall well written and easy to follow in most parts. The experiments are extensive.

**Weaknesses:**

I would not claim these as weaknesses, but suggestions that might help improve the work.

First, while it is good to see that the method shows improvements, I would suggest not characterizing the results as "a such huge improvement". On one hand, a 2%-3% improvement is not huge. On the other hand, the improvements just meet the expectation since more data are used for training. In addition to show quantitative results, it is essential to report some other metrics, such as training time, to render a more comprehensive comparison.

Second, as discussed in Sec. 2, there are other strategies for handling heterogeneous labels in multi-dataset joint training, and I am curious whether existing strategies can be directly used for video instance segmentation. If yes, how about the performance?

Third, in terms of formulation,  E_\infty is used in Sec 3.2, but does not appear in Eqs. 2-5. It is important to ensure consistency. Eq. 6 is not properly formulated as well; a same symbol is used for both input and output, i.e., $X_{l-1}$.

**Questions:**

Beyond the questions in [weaknesses], I have one last question: is the model applicable to open-world scenarios? or at least can it generalize to other datasets that not involve in model training?


**Limitations:**

Authors tried to discuss limitations; however, the discussion provided is rather elementary and does not offer crucial insights into the method. More in-depth analysis should be provided.

---

> ### Author Rebuttal · Authors · 2023-08-10
>
> At the beginning, we want to thank you for the detailed, insightful and constructive comments.
>
> ### More metrics
>
> Thanks for your careful suggestions. Firstly, it’s true that performance will improve as data volume increases, but our method successfully alleviates the heterogeneity in category space of multiple VIS datasets, which leads to a better improvement than simply training data together. As for the characterization of "such a huge improvement", we will carefully revise them in the final version.
>
> We also tested the training time and model parameters with some current methods. We list the model parameters and FPS of SeqFormer[1] (220M/27.7), VITA[2] (247M/22.8), and our TMT-VIS (250M/21.4). Also, we compared our results with the previous multi-datasets joint training method UNINEXT[3], and the details are shown in Table 2-1.
>
> Table 2-1. Comparison with UNINEXT on YTVIS-19.
>
> | Method        | Datasets Number | Training time  | Backbone   | AP   |
> | ------------- | :-------------: | -------------- | :--------: | ---- |
> | UNINEXT       | 8               | 3 days         | ConvNext-L | 64.3 |
> | TMT-VIS(Ours) | 4               | 1 day 12 hours | Swin-L     | 64.9 |
>
> ### Transfer multi-datasets joint training methods to VIS.
>
> We transfer existing image-level strategies on handling heterogeneous labels in multi-dataset joint training to VIS tasks based on Mask2Former-VIS, and results are shown in the table. We integrate the popular multiple-dataset object detection method, UniDet[4], and compare it to our design. UniDet firstly trains a single partitioned detector on multiple datasets with shared backbone, dataset-specific outputs and loss and then unifies the outputs of the partitioned detector in a common taxonomy completely automatically. Another popular multiple-dataset object detection method is OmDet[5], but it imposes restrictions on its own specific architecture. As the table shows, UniDet has a lower performance than our design (3.7 AP lower), given that it trains in two steps.
>
> Table 2-2. Experiments on popular multi-datasets joint training methods.
> | Method        | Backbone  | AP   |
> | ------------- | --------- | ---- |
> | UniDet        | ResNet-50 | 48.9 |
> | TMT-VIS(Ours) | ResNet-50 | 52.6 |
>
>
> ### Formulation.
> Thanks again for your careful suggestions. $E_{\infty}$ is actually corresponding to the $E_{1}$ in the Eqs.2-5, but we misuse the \mathcal grammar to $E_{1}$ so it turns out to be $E_{\infty}$. As for the X_{l-1}, in the final version we will add a ‘\prime’ to distinguish between the input and output of the different layers of TCM.  We will revise the whole paper thoroughly and with more carefulness to ensure both consistency and properness of formulations.
>
> ### Open-world scenarios or other video datasets
>
> We update and reorganize Tab.3 in the original submitted paper, and the results are shown in the rebuttal pdf. The table contains the zero-shot performance of TMT-VIS. The results show the potentials of our model to open-world settings.
>
> We also tested our method on VIPSeg[6] based on the popular Video K-Net[7], the results are shown in Table 2-3. VIPSeg is a large-scale dataset for video panoptic segmentation (VPS)[8], a task which aims to simultaneously predict object classes, bounding boxes, masks, instance id associations, and semantic segmentation in video frames. There are a total of 3,536 videos with 84,750 pixel-wise annotated frames in VIPSeg, which covers 232 scenarios with 124 categories, including 58 things’ classes and 66 stuff’s classes. Here, VPQ and STQ are both metrics from the VPS task. With our additional design, we compiled taxonomic information and injected them to thing and stuff kernels, and trained YTVIS and VIPSeg jointly, with the total training epochs set to 12. The results are shown below, with our design, the performance of Video K-Net increases with a 2.3% VPQ improvement and 2.6% STQ improvement. This further demonstrates the transferability of our design in in-the-wild scenarios.
>
> Table 2-3. Results on VIPSeg Dataset.
> | Method      | Multiple Dataset? | Backbone  | VPQ  | STQ  |
> | ----------- | :---------------: | --------- | ---- | ---- |
> | Video K-Net |                   | ResNet-50 | 26.1 | 33.1 |
> | Video K-Net | √                 | ResNet-50 | 27.6 | 34.8 |
> | TMT-VIS     | √                 | ResNet-50 | 28.4 | 35.7 |
>
> ### Reference
>
> [1] Wu J, Jiang Y, Bai S, et al. Seqformer: Sequential transformer for video instance segmentation[C]//European Conference on Computer Vision. Cham: Springer Nature Switzerland, 2022: 553-569.
>
> [2] Heo M, Hwang S, Oh S W, et al. Vita: Video instance segmentation via object token association[J]. Advances in Neural Information Processing Systems, 2022, 35: 23109-23120.
>
> [3] Yan B, Jiang Y, Wu J, et al. Universal instance perception as object discovery and retrieval[C]//Proceedings of the IEEE/CVF Conference on Computer Vision and Pattern Recognition. 2023: 15325-15336.
>
> [4] Zhou X, Koltun V, Krähenbühl P. Simple multi-dataset detection[C]//Proceedings of the IEEE/CVF Conference on Computer Vision and Pattern Recognition. 2022: 7571-7580.
>
> [5] Zhao T, Liu P, Lu X, et al. Omdet: Language-aware object detection with large-scale vision-language multi-dataset pre-training[J]. arXiv preprint arXiv:2209.05946, 2022.
>
> [6] Miao J, Wang X, Wu Y, et al. Large-scale video panoptic segmentation in the wild: A benchmark[C]//Proceedings of the IEEE/CVF Conference on Computer Vision and Pattern Recognition. 2022: 21033-21043.
>
> [7] Li X, Zhang W, Pang J, et al. Video k-net: A simple, strong, and unified baseline for video segmentation[C]//Proceedings of the IEEE/CVF Conference on Computer Vision and Pattern Recognition. 2022: 18847-18857.
>
> [8] Kim D, Woo S, Lee J Y, et al. Video panoptic segmentation[C]//Proceedings of the IEEE/CVF Conference on Computer Vision and Pattern Recognition. 2020: 9859-9868.

---

> > ### Comment · Reviewer_wd7m · 2023-08-16
> > **thanks for the response**
> >
> > Thanks for the detailed rebuttal and new experiments. Most of my concerns have been properly addressed. But I find Table 2-1 to be somewhat perplexing since the dataset numbers and underlying backbones are different. This makes it hard to make a proper comparison.
> >
> > In addition, I read other reviewers' comments and would like to discuss more following Reviewer 5w4z's `why video?` question. On one hand, the model shows favorable performance in the new `image` setting, i.e.,  a gain of +4% is undoubtedly promising. But does this outcome imply that the method's uniqueness is not exclusively tied to `video`? On the other hand, the newly added experiments coutine to rely on YTVIS-19. I am actually curious about results on datasets exclusively comprised of images like COCO and LVIS. While I don't anticipate fresh results at current stage, the question seems to be remain unanswered and I expect authors give more insights.

---

> > > ### Author Response · Authors · 2023-08-20
> > >
> > > We are sincerely grateful to you for the precious time and selfless efforts you have devoted to reviewing and commenting our paper.
> > >
> > > Our research is primarily anchored on Video Instance Segmentation (VIS), not image instance segmentation, and we are willing to accept all types of results—be it advancements or drawbacks—stemming from jointly training image datasets. We are exploring the usage of joint training methods in VIS training with a view to combine existing standalone and specific VIS datasets into a more exhaustive resource.
> > >
> > > Our motivation stems from the notable size difference between current image instance segmentation datasets and VIS datasets. As an example, LVIS hosts 160k images together with 2M instance mask annotations, thereby providing a richer variety of scenarios compared to VIS datasets. Conversely, VIS datasets are typically smaller and the resources available to us are largely fragmented, field-specific datasets rather than larger, established ones. Given that there's no prevalent joint training strategy for VIS and a lack of widely recognized large-scale VIS datasets, our proposition is to jointly train the available VIS datasets to enhance overall VIS performance.
> > >
> > > Table 1 Key statistics of Image Instance Segmentation datasets
> > >
> > > |            | COCO | LVIS |
> > > | ---------- | ---- | ---- |
> > > | Images     | 164K | 160K |
> > > | Categories | 80   | 1203 |
> > > | Masks      | 1.2M | 2M   |
> > >
> > > Table 2. Key statistics of multiple VIS datasets.
> > >
> > > |                   | YT19 | YT21 | OVIS | UVO    |
> > > | ----------------- | ---- | ---- | ---- | ------ |
> > > | Videos            | 2883 | 3859 | 901  | 11228  |
> > > | Categories        | 40   | 40   | 25   | 81     |
> > > | Instances         | 4883 | 8171 | 5223 | 104898 |
> > > | Masks             | 131k | 232k | 296k | 593k   |
> > > | Masks per Frame   | 1.7  | 2    | 4.7  | 12.3   |
> > > | Objects per Video | 1.6  | 2.1  | 5.8  | 9.3    |

---

### Official Review · Reviewer_5w4z · 2023-07-09

**Soundness:** 1 poor
**Presentation:** 1 poor
**Contribution:** 1 poor
**Rating:** 6
**Confidence:** 5

**Summary:**

In this paper, the authors propose a multi-dataset joint training method for the video instance segmentation task. They build on the MaskFormer-VIS model, introducing two additional modules: the Taxonomy Compilation Module and the Taxonomy Injection Module. And the proposed method shows compelling performance.

**Strengths:**

S1. This paper tackles an overlooked problem: multi-dataset joint training for the video instance segmentation task.

**Weaknesses:**

W1. A significant concern I have is with the explicit relevance and motivation of the proposed methodologies for the task of "video" instance segmentation. The paper currently lacks clear justification as to why these particular methods are designed and show impressive performance specifically for this task. It would benefit from a more in-depth exploration and justification of why these methodologies are uniquely suited for "video" instance segmentation.

W2. The paper needs an overall improvement in writing quality. The related works section contains several inaccuracies that need rectification. To specify:
  - L83: The claim that Mask R-CNN [13] and MinVIS [18] use a tracking branch is incorrect.
  - L89: It is incorrect to label [47, 28] as MOTS works; they are, in fact, MOT methods.
  - L90: The assertion that IDOL [39] is based on GenVIS [15] is inaccurate.

Additionally, there's a misleading terminology used for the proposed module on L296; it should be the Taxonomy Compilation Module (TCM), not the Taxonomy Extraction Module (TEM).

W3. The paper seems to possess marginal novelty. In particular, as stated in L116-117, the proposed method leverages taxonomic embedding (as I understand, these are embeddings from the VLM), rather than language embeddings. The distinction of this work compared to previous multi-dataset joint training methods isn't clear. It would be advantageous if the authors could provide a more explicit elaboration on the unique novelty of their work.

**Questions:**

Major concerns are raised in the weakness section.

**Limitations:**

This paper has discussed its limitations and societal impact in the main paper.

---

> ### Author Rebuttal · Authors · 2023-08-10
>
> At the beginning, we want to thank you for the detailed, insightful and constructive comments.
>
> ### Uniqueness for "video" instance segmentation.
>
> When injecting the taxonomic embeddings, we add a spatio-temporal adapter to generate video-specific modulated taxonomic embeddings. This approach is not only parameter-efficient but also adapts the output text embeddings from text encoder to fit the scenarios of multiple frames. Thus, the modulated taxonomic embeddings are able to interact with the frame features across temporal dimensions through cross-attention and FFN operations. Since the difficulty of VIS tasks lies in segmenting while modeling the trajectories, which grows in polynomials along with the number of video frames, simply using queries from transformer decoder to gradually refine can be troublesome and ineffective. On the other hand, injecting modulated taxonomic embeddings as guidance can provide a prior for transformer queries, and because the modulated embeddings are filtered to possess the most possible categories’ semantic information, such injection can help queries to converge to the desired instances faster and model instances' trajectories more precisely.
>
> When we downsample the YTVIS video clips to only 1 frame per video, the video instance segmentation has shrunk back to image instance segmentation.  we could notice that our design provides less improvement in such an image-level setting as with higher fps. The results are reasonable since the spatio-temporal adapter contracts to provide only spatial adjustment. With the backbone trained on COCO images, the strong segmentation backbone provides a great segmentation capacity in image-level, making the final results higher than training on the original YTVIS-19. This further illustrates that our design is specifically designed for "video" instance segmentation.
>
> Table 1-1. Experiments on different frames per video.
> | Method      | Testing Dataset | Frames Per Video | AP   |
> | ----------- | :-------------: | :--------------: | ---- |
> | Mask2Former | YTVIS-19         | 1                | 49.2 |
> | TMT-VIS     | YTVIS-19         | 1                | 53.5 |
> | Mask2Former | YTVIS-19         | ~30              | 46.4 |
> | TMT-VIS     | YTVIS-19         | ~30              | 52.6 |
>
> ## Overall writing
>
> Thanks for your careful suggestions about the misplacement of citations in related works as well as the terminologies of the `Taxonomy compilation module'. We will revise the whole paper thoroughly and with more carefulness.
>
> To clarify some of the mistakes in our proposed version: MaskTrack R-CNN is the baseline method of online VIS method, which is built by embedding a simple tracking branch to Mask R-CNN. MinVIS implements a strong query-based image instance segmentation model (Mask2Former) on individual frames and associate query embeddings by bipartite matching. Based on Deformable-DETR, IDOL introduces a contrastive learning head that acquires discriminative instance embeddings for association. And as for the MOTS method mentioned in related works, the baseline method is TrackR-CNN. It extends the Mask R-CNN with three-dimensional convolution to combine contextual information and deploys association head to extract instance embedding for data association. Another important SOTA method is PointTrack, which performed the tracking-by-instance segmentation
> paradigm. It first obtains high-quality instance segmentation results with spatial embedding, and then extracts instance features from the
> segmentation results. As for the misuse of terminologies, we want to again deeply apologize for the carelessness.
>
> ### Novelty of TMT-VIS.
>
> A major challenge in multiple-dataset joint training is the heterogeneity of multiple datasets’ category space. As a result of the difference in category space, though mask precision increases with the data volume, dataset biases might hinder models from generalization: simply utilizing multiple datasets will dilute the attention of models on different categories. Therefore, increasing the data scale and enriching label space while improving classification precision become a huge challenge for researchers. This phenomenon was shown in our ablation study.
>
> The UNINEXT[1] is the first DETR-based method which jointly trains multiple VIS datasets. It simply utilizes BERT language encoder to generate language embeddings of categories from all video datasets, and fuse the information with visual embeddings through a simple bi-directional cross-attention module. However, UNINEXT has no video-specific design, and it doesn't use the language embeddings to predict a set of possible set of categories, so the semantic information of VIS categories are simply aggregated without further operations.  Our method, on the other hand, adjusts the language embeddings via a taxonomy compilation module which consists of a spatio-temporal adapter and a FFN network, as well as of a taxonomy injection module which fuses the taxonomy information into video queries with a multi-level cross-attention and self-attention layers. In the taxonomy compilation module, taxonomy embeddings are designed to interact with video features in order to unearth the potential taxonomy contained in each of the video frames. After this, we calculated the dot product between the different modulated taxonomic embeddings to predict the most relevant taxonomy in the given video. By compiling and injecting the possible taxonomic information to queries as guidance, queries  tend to converge to the desired instances faster and finally come up with better precision.
>
>
> ### Reference
> [1] Yan B, Jiang Y, Wu J, et al. Universal instance perception as object discovery and retrieval[C]//Proceedings of the IEEE/CVF Conference on Computer Vision and Pattern Recognition. 2023: 15325-15336.

---

> > ### Author Response · Authors · 2023-08-20
> >
> > Dear Reviewer 5w4z,
> >
> > We are sincerely grateful to you for the precious time and selfless efforts you have devoted to reviewing our paper.
> >
> > Since the deadline of discussion is approaching, we would like to inquire whether our response has addressed your concerns and if you have the time to provide further feedback on our rebuttal. We are more than willing to engage in further discussion.
> >
> > Best regards,
> > Authors of paper 11160

---

> > > ### Comment · Reviewer_5w4z · 2023-08-20
> > >
> > > Thank you for the authors' response. I have carefully reviewed the feedback from other reviewers and the authors' comments on them. In particular, both Reviewer k3N8 and Reviewer wd7m shared my initial question regarding the choice of "why video?". I've read the authors' explanations on this matter.
> > >
> > > I understand that video instance segmentation datasets tend to be smaller than image datasets, and that joint training for a large vocabulary presents its own set of challenges. My primary concern, however, centers on the unique benefits the proposed method offers for video. As far as I know, M2F-VIS uses 2 frames from a single video during the VIS task training. Have the authors followed this implementation setting? Authors mentioned integrating a 'spatio-temporal adapter to generate video-specific modulated taxonomic embeddings.' I wonder if using just two frames can effectively capture video-oriented knowledge for the model and assess its efficacy.

---

> > > > ### Author Response · Authors · 2023-08-21
> > > >
> > > > We express our sincere gratitude to you for generously dedicating your valuable time and selfless efforts to thoroughly review and provide insightful comments on our paper.
> > > >
> > > > As shown in previous ablations in submitted paper and rebuttal, we show that our strategy is plug-and-use to popular VIS methods across VIS datasets. In experiments on M2F-VIS, we follow the original settings to set the input video frames per video to 2. While in experiments based upon VITA, when input video frames per video is set to 6, the overall performance becomes SOTA with our design. The effectiveness of capturing video-oriented knowledge with different input frames in the design of TCM is a problem worth discussion, but is not the main focus of our current paper.
> > > >
> > > > In M2F-VIS setting, we previously experimented that M2F-VIS tends to have a drop in performance when input frame number grows larger than 2 (in Table 1). This indicates that 2 frames are a basic module for M2F-VIS to build up temporal correlations. Even though with more input frames comes more features, the complexity of modeling instance trajectories, on the other hand, hinders the performance. In such settings, 2 is the most effective choice of parameter. As for capturing video-oriented knowledge with more input frames, we haven't fully experimented on this, which is not the main focus of our work.
> > > >
> > > > Table 1. M2F-VIS performance on YTVIS-19 on R-50 backbone
> > > > |         | 1    | 2    | 3    | 4    | 6    |
> > > > | ------- | :--: | :--: | :--: | :--: | :--: |
> > > > | M2F-VIS | 45.1 | 46.4 | 46.0 | 45.7 | 45.6 |

---

> > > > > ### Comment · Reviewer_5w4z · 2023-08-21
> > > > >
> > > > > Thank you for the comments! I concur that the current M2F-VIS design isn't ideally suited for capturing temporal relationships, and its end-to-end nature might substantially increase resource demands. However, with the use of multiple frames as input and observing the results from VITA, which showcases SOTA performance, my reservations have been alleviated. As a result, I'd like to amend my rating to "weak accept". I hope the authors delve deeper into these concerns and provide further clarifications in their revised paper.

---

> > > > > > ### Comment · Reviewer_5w4z · 2023-08-21
> > > > > >
> > > > > > Also, taking into account the responses provided by the authors above, I believe that ablation studies using VIS methods other than M2F-VIS would be greatly beneficial for readers.

---

> > > > > > > ### Author Response · Authors · 2023-08-21
> > > > > > >
> > > > > > > Dear Reviewer 5w4z,
> > > > > > >
> > > > > > > We greatly appreciate your acknowledgement of our rebuttal and your appreciation of our work. We sincerely thank you for your valuable feedback and support. In accordance with your suggestions, we will diligently refine the paper and address any confusing sections. Your constructive feedback will contribute to enhancing the overall quality of our work. Once again, thank you for your invaluable contribution.
> > > > > > >
> > > > > > > Best regards,
> > > > > > > Authors of paper 11160.

---

> > > > > > > > ### Comment · Reviewer_5w4z · 2023-08-21
> > > > > > > >
> > > > > > > > Finally, I'd like to draw attention to the related work section. For expert readers, such as reviewers, having misleading content in this section can significantly diminish the credibility of the paper. While the authors have acknowledged and made some revisions, I urge them to further review and refine this section carefully. It's essential for the authors to take responsibility for their content, especially since many subsequent studies reference the related work sections of top-conference papers.

---

> > ### Comment · Area_Chair_kcB3 · 2023-08-20
> > **Response Required**
> >
> > Dear Reviewer 5w4z,
> >
> > Given you are in the minority with a reject it is critical that you read the rebuttal and inquire about any outstanding issues you feel are still not addressed.
> >
> > Thank you,
> > AC

---

### Author Rebuttal · Authors · 2023-08-10

At the beginning, we want to thank all reviewers for the detailed, insightful and constructive comments.

In the PDF file, we have attached the reorganized version of the ablation study on training with multiple VIS datasets, as well as the key statistics of these datasets. In this part, we want to address the novelty-related and the robustness-related concerns from reviewers.

### Novelty

A major challenge in multiple-dataset joint training is the heterogeneity of multiple datasets’ category space. As a result of the difference in category space, though mask precision increases with the data volume, dataset biases might hinder models from generalization: simply utilizing multiple datasets will dilute the attention of models on different categories. Therefore, increasing the data scale and enriching label space while improving classification precision become a huge challenge for researchers. This phenomenon was shown in our ablation study.

A straightforward method of enhancing classification precision is to aggregate category-related language embeddings into the modelling process, and the aggregation targets are queries to be more specific in DETR-based models. Previous multiple-dataset joint training methods, such as UNINEXT[1], adopt this idea and manage to fuse category-guided prompt with video features. However, aggregating all category-guided prompts introduces irrelevant semantic information. We argue that the total instances in an input video are far less than the size of the whole label space, and so that by filtering out some irrelevant categories, the aggregation of language embeddings becomes more effective and efficient. Thus, in our TCM, taxonomy embeddings are designed to interact with video features in order to unearth the potential taxonomy contained in each of the video frames. After this, we calculated the dot product between the different modulated taxonomic embeddings to predict the most relevant taxonomy in the given video. By aggregating these most relevant compiled taxonomic embeddings into query features though cross-attention and self-attention modules, we could have refined query features with not only richer but also condensed taxonomy information, in contrast with previous methods, which further improves the performance.


### Robustness of our method across various datasets

It's true that there are degradation when jointly training UVO and YTVIS-19, but we argue that in most of the training settings TMT-VIS improves significantly better than Mask2Former (see the table in rebuttal PDF), and so this result can't indicate that our method lacks robustness. In the following, we try to explain the reason why this may occur.

The drop in performance may be caused by the imbalance in scale among VIS datasets. YTVIS-19 takes advantage of an existing dataset called YouTube-VOS.  OVIS is collected specifically for video instance segmentation in occluded scenes. UVO, on the other hand, adopts videos from Kinetics-400, which are human-centric and contain diverse sets of human actions and human-object interactions, and UVO is densely annotated with many more instances.  The key statistics of these datasets are shown in the following table, which corresponds to the Table.1 of the supplementary file.  As illustrated in the table of key statistics of datasets, the imbalance between these VIS datasets is significant. When jointly training YTVIS-19 with UVO, the model's parameters converge to fit UVO, which leads to the degradation of performance on validation on YTVIS-19. When jointly training YTVIS-19 and UVO with OVIS, the imbalance is alleviated, and results in a notable performance improvement.

It’s worth noting that due to the scale of VIS datasets, especially YTVIS-19, the final results usually have a fluctuation of approximately 0.3AP, and this may also be the direct reason for the seeming drop of our TMT-VIS compared with Mask2Former-VIS because in other training settings our TMT-VIS significantly outperforms Mask2Former-VIS.

#### Reference
[1] Yan B, Jiang Y, Wu J, et al. Universal instance perception as object discovery and retrieval[C]//Proceedings of the IEEE/CVF Conference on Computer Vision and Pattern Recognition. 2023: 15325-15336.

---

### Decision · Program_Chairs · 2023-09-21

**Decision:**

Accept (poster)

**Comment:**

The AC and reviewers appreciate the authors rebuttal and discussion. All reviews, rebuttals and comments were carefully read and considered by the AC. All reviewers appreciated the idea of multi-dataset training but were not convinced that the methodology was specific to video. There were also concerns with the writing of the paper and missing experiments. The authors provided comprehensive rebuttals, filling in many of the gaps in the experimental section. 4 of 5 reviewers were positive in the end and thus the AC has decided to accept the paper. Please make the promised modification to the camera ready version of the paper.